# Time Series Kernels based on
# Nonlinear Vector AutoRegressive Delay Embeddings

**Giovanni De Felice**    **John Y. Goulermas**[*]    **Vladimir V. Gusev**
Department of Computer Science
University of Liverpool
gdefe@liverpool.ac.uk    gusev@liverpool.ac.uk

## Abstract

Kernel design is a pivotal but challenging aspect of time series analysis, especially in the context of small datasets. In recent years, Reservoir Computing (RC) has emerged as a powerful tool to compare time series based on the underlying dynamics of the generating process rather than the observed data. However, the performance of RC highly depends on the hyperparameter setting, which is hard to interpret and costly to optimize because of the recurrent nature of RC. Here, we present a new kernel for time series based on the recently established equivalence between reservoir dynamics and Nonlinear Vector AutoRegressive (NVAR) processes. The kernel is non-recurrent and depends on a small set of meaningful hyperparameters, for which we suggest an effective heuristic. We demonstrate excellent performance on a wide range of real-world classification tasks, both in terms of accuracy and speed. This further advances the understanding of RC representation learning models and extends the typical use of the NVAR framework to kernel design and representation of real-world time series data.

## 1 Introduction

Time series are arguably one of the most important types of data in the modern era (Hamilton, 2020) and ubiquitous both in scientific research (Strogatz, 2018) and practical applications (Zhang et al., 2018; Zeroual et al., 2020). A key element in the design of most machine learning protocols for time series is a quantification of similarity (Ding et al., 2008; Abanda et al., 2019; Echihabi et al., 2020). This is especially true for kernel methods (Schölkopf et al., 2001), which search for linear solutions after projecting the data into a higher (possibly infinite) dimensional space and represent an effective alternative to non-linear models. The performance of kernel methods is heavily impacted by the definition of a positive semi-definite (PSD) similarity function for the type of data at hand, i.e., a *kernel*. However, their design is challenging for structured data such as univariate (UTS) and multivariate time series (MTS), which exhibit temporal autocorrelation, inter-attributes (or dimensions) dependencies and possibly a variety of temporal distortions and misalignments (Paparrizos et al., 2020).

To overcome this, a promising approach is to identify and compare the underlying dynamics rather than the raw observed data. Notably, kernels based on Reservoir Computing (RC) stand out in this context (Chen et al., 2013; Bianchi et al., 2020). RC-based kernels use a randomized and untrained layer of recurrently connected nodes (the *reservoir*) to map each time series into a rich dynamical feature space. The similarity is then obtained by comparing individual *readouts* capturing dynamics. In practice, RC is highly sensitive to a large set of hyperparameters having little interpretability, the optimization of which is challenging for the comprehensive range of values.

---

[*]Deceased May 2022

37th Conference on Neural Information Processing Systems (NeurIPS 2023).

Recently, a theoretical work by Bollt (2021) has demonstrated that a simple RC can be formally rewritten as a non-linear vector autoregressive (NVAR) model. In the NVAR framework, the *reservoir* is replaced by a simple concatenation of the input series with time-delayed copies and nonlinear functionals, such as products (Gauthier et al., 2021). This reduces the number of hyperparameters and has proven to be very effective in chaotic systems forecasting (Shahi et al., 2022), as much to earn the title of "Next-Generation Reservoir Computing". However, its applicability and performance for real-world data and outside of forecasting dynamical systems are largely unexplored.

In this work, we investigate whether an NVAR process can replace a reservoir for kernel design and if the extracted features are just as effective in comparing real-world time series data. We propose a very efficient and effective NVAR kernel for UTS and MTS data. The main idea is as follows (Fig. 1). First, we enrich each time series with lagged copies of itself and additional nonlinear terms, forming a high-dimensional NVAR embedding. Then, we extract a linear parameterization of the time evolution in the embedding space, representative of the underlying dynamics, analogously to RC representation learning methods (Bianchi et al., 2020). Finally, these fitted parameters are used to compute the similarity between the time series. Our main contributions branch out into different communities investigating time series, namely, kernel design, RC and NVAR, bringing specific benefits to each of them:

- From a kernel design perspective, we provide an NVAR kernel for UTS and MTS and a general parameter setting based on simple heuristics. Based on our experiments on a wide range of datasets, the proposed approach matches the state-of-the-art (SOTA) in terms of accuracy and is considerably faster than SOTA, representing the best compromise between accuracy and efficiency.

- From the perspective of RC, we advance its success for TS representation by incorporating NVAR-made dynamics. The resulting model is more accurate, interpretable, and non-recurrent, which overcomes difficulties in hyperparameter optimization and sheds light on the potential RC performance.

- From an NVAR perspective, we extend the framework to real-world time series representation and kernels, well beyond the original work on forecasting synthetic, noise-free, and chaotic dynamical systems (Gauthier et al., 2021).

- We establish a connection to Takens' theorem (Takens, 1981) and the field of *state space reconstruction* (Sauer et al., 1991), which theoretically underpin our approach, highlighting a compelling avenue for the use of dynamical systems theory in machine learning.

## 1.1 Notation

Through the paper, we indicate variables as lowercase ($x$); constants as uppercase ($X$); vectors and UTS as bold lowercase ($\mathbf{x}$); matrices and MTS as bold uppercase $\mathbf{X}$. An index between square parenthesis $x[n]$ indicates the $n$-th sample of a set. For an MTS $\mathbf{X}$, the corresponding lower-case $\mathbf{x}_t$ indicates all dimensions at time stamp $t$ while $\mathbf{x}^d$ indicates dimension $d$ at all timestamps.

## 2 Related methods

In this section, we provide a background on the previously proposed kernels for time series data, grouped by category. Despite the abundance of time series similarity measures (Yang & Shahabi, 2004; Paparrizos & Gravano, 2015; Janati et al., 2020), we limit the discussion to PSD metrics.

**Lock-step** Lock-step approaches view time series as static vectors and directly use common distance metrics (Cha, 2007). The PSD property is usually obtained by superposing a linear or radial kernel (Schölkopf et al., 2002). This is very efficient, but ignores any temporal structure within the series and is not applicable for different lengths.

**Elastic** Elastic measures account for time distortions (e.g. shifts) or different lengths between sequences. Lu et al. (2008) proposed to interpolate the series and treat the problem as a distance between curves. The Global Alignment Kernel (GAK) (Cuturi, 2011) and KDTW (Marteau & Gibet, 2014) instead are built on the well-known Dynamic Time Warping (DTW) (Berndt & Clifford, 1994) and compute the similarity from the cost of one-to-many alignments between the series. The

Shift-invariant kernel (SINK) (Paparrizos & Franklin, 2019) computes a cross-correlation similarity in the Fourier domain. However, for the aforementioned methods, the computation of the pairwise similarity is computationally expensive and does not account for the relations between different attributes in the multivariate case.

**Model-based**  Model-based kernels process the series with a probabilistic or deterministic model and base the similarity on the extracted information. Such a model can be a single generative model, as in the Fisher kernel (Jaakkola et al., 1999) or the probability product kernels (Jebara et al., 2004), or a parameterized family of probability distributions, as in the autoregressive kernel (Cuturi & Doucet, 2011). Similarly to the latter, the Time Cluster Kernel (TCK) (Mikalsen et al., 2018) is obtained from an ensemble of Gaussian mixture posteriors, sharing the same parametric form but trained on different subsets of the dataset. Among non-probabilistic approaches, the Learned Pattern Similarity (LPS) (Baydogan & Runger, 2016) uses an ensemble of regression trees to extract local patterns from each series. The downside of all such methods comes from the specific functional form, which may limit the generalizability of the extracted features. Ensemble methods are also unsuited for datasets with few training samples.

**Reservoir-based**  Reservoir Computing (RC) is a class of recurrent neural networks that keep recurrent connections untrained in order to overcome the high cost of back-propagation through time (Werbos, 1990) and the vulnerability to exploding and vanishing gradient (Pascanu et al., 2013). Its simplest form, the Echo State Network (ESN) (Jaeger, 2001), is composed of three layers: an *input layer*, a hidden layer of connected neurons (*reservoir*) and a *readout layer*:

$$\text{input:} \qquad \mathbf{u}_t = \mathbf{W}^{in}\mathbf{x}_t \tag{1}$$

$$\text{reservoir:} \qquad \mathbf{r}_t = (1 - \alpha)\mathbf{r}_{t-1} + \alpha f(\mathbf{A}\mathbf{r}_{t-1} + \mathbf{u}_t + \epsilon_t) \tag{2}$$

$$\text{readout:} \qquad \mathbf{y}_t = \mathbf{W}^{out}\mathbf{r}_t + \mathbf{c} \tag{3}$$

where $\mathbf{x}_t \in \mathbb{R}^{d_x}$, $\mathbf{u}_t \in \mathbb{R}^{d_r}$ and $\mathbf{r}_t \in \mathbb{R}^{d_r}$ are, respectively, the input, its upscaling projection and the reservoir states at time t (with $d_r \gg d_x$ and $\mathbf{r}_0 := \mathbf{0}$); $0 \leq \alpha \leq 1$ is the leak parameter; $\epsilon_t$ is an additive noise and $f$ is any non-linear activation function. The reservoir ($\mathbf{A} \in \mathbb{R}^{d_r \times d_r}$) and the projection ($\mathbf{W}^{in} \in \mathbb{R}^{d_r \times d_x}$) matrices are kept untrained, while the readout ($\mathbf{W}^{out} \in \mathbb{R}^{d_y \times d_r}$) and the bias term ($\mathbf{c}$) are learned by fitting the reservoir states to the output ($\mathbf{y_t} \in \mathbb{R}^{d_y}$) via ridge regression.

Despite their randomness, the reservoir states constitute a rich pool of heterogeneous features that give complete knowledge of the underlying dynamical system generating the observed time series (Løkse et al., 2017; Bianchi et al., 2020). In fact, as opposed to fitting a specific functional form, a solid theoretical ground ensures the quality of the reservoir features (Hart et al., 2020; Gonon et al., 2023; Gonon & Ortega, 2021; Hart et al., 2021). Nevertheless, performance is highly sensitive to the choice of a large set of hyperparameters controlling the reservoir initialization and update (Lukoševičius, 2012). For instance, *input scaling* ($\|\mathbf{W}^{in}\|$) and *spectral radius* ($\rho(\mathbf{A})$) assure stable reservoir dynamics (Gallicchio, 2019) and control the degree of injected nonlinearity. This is not easy to judge and requires experienced insight into nonlinear dynamics. In general, the interpretation and setting of ESN parameters is a complex task and, to this day, an active area of research (Dong et al., 2022; Steiner et al., 2022; Zhang et al., 2022). Simple supervised optimization, such as cross-validation (CV), is often prohibitive, as the optimization space is large and the recursive nature of the reservoir forces the computation of Eq. 2 for the whole length of the series before assessing the performance of one hyperparameter setting.

Regarding RC-based kernels, Chatzis & Demiris (2011) originally built a radial kernel directly on top of the reservoir states. In Chen et al. (2013), all series are processed by a shared reservoir and compared based on the respective readouts, which are individually trained on the one-step-ahead prediction task ($\mathbf{y}_t = \mathbf{x}_{t+1}$). Finally, Bianchi et al. (2020) observed that more information is retained by using $\mathbf{y}_t = \mathbf{r}_{t+1}$, and an effective kernel can be obtained by applying a radial function on the vectorization of all the readout weights.

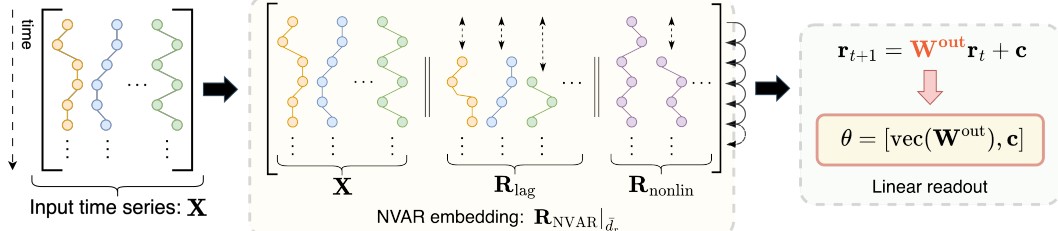

Figure 1: Building of each representation vector in the NVAR kernel. The input series $\mathbf{X}$ is concatenated with $\mathbf{R}_{lag}$, which contains $k$ lags of all input dimensions, and with $\mathbf{R}_{nonlin}$, containing products between dimensions of $\mathbf{X}$ and $\mathbf{R}_{lag}$. Among all possible concatenation terms, only a random subset of dimensions is considered. The readout then extracts the linear dynamics ($\mathbf{W}^{out}$) of the embedding states and the bias term ($\mathbf{c}$) by performing a ridge regression fit along the temporal dimension. These fit parameters are vectorized to form the representation.

## 3 Methodology

This section presents the main building blocks of NVAR models, followed by our proposal of an NVAR-based kernel and a general setting for the hyperparameters that govern it.

### 3.1 The Nonlinear Vector AutoRegressive model

Recently, the work of Bollt (2021) has established a connection between the ESN architecture and Nonlinear Vector AutoRegressive (NVAR) models. Under specific conditions, this takes the form of a formal equivalence, where the coefficients of an NVAR process can be expressed in terms of $\mathbf{A}$, $\mathbf{W}^{in}$ and $\mathbf{W}^{out}$. By building on such equivalence, in Gauthier et al. (2021), the input layer (Eq. 1) and the reservoir (Eq. 2) of an ordinary ESN are replaced with the deterministic operations that lead to a feature matrix $\mathbf{R}_{NVAR}$. In detail, this is the result of concatenating three terms: the input series ($\mathbf{X}$), $k$ lagged copies of the input, each of which is increasingly spaced by $s$ time-stamps ($\mathbf{R}_{lag}$), and all possible products[2] between lagged and unlagged dimensions, up to a polynomial order $n$ ($\mathbf{R}_{nonlin}$):

$$\mathbf{r}_{lag,t} = \begin{bmatrix} \mathbf{x}_{t-s} \,||\, \mathbf{x}_{t-2s} \,||\, ... \,||\, \mathbf{x}_{t-ks} \end{bmatrix}$$

$$\mathbf{r}_{nonlin,t} = \begin{bmatrix} \mathbf{x}_t \,||\, \mathbf{r}_{lag,t} \end{bmatrix} \lceil\otimes\rceil \begin{bmatrix} \mathbf{x}_t \,||\, \mathbf{r}_{lag,t} \end{bmatrix} \lceil\otimes\rceil \, ... \quad (n \text{ times})$$ 

$$\mathbf{R}_{NVAR} = \mathbf{X} \,||\, \mathbf{R}_{lag} \,||\, \mathbf{R}_{nonlin}$$

(4)

where $\cdot\,||\,\cdot$ is the column-wise concatenation and $\cdot\lceil\otimes\rceil\cdot$ performs the outer product and concatenates all unique monomials.

This procedure is very efficient as it is non-recurrent and the size of $\mathbf{R}_{NVAR}$ is usually smaller than a typical reservoir. On top of that, it is determined by a small set of integer hyperparameters: $k$, $s$ and $n$. In principle, to sustain the equivalence to RC, an infinite amount of lags ($k \to \infty$) should be considered. Nevertheless, it has been observed that a small $k$ with $n = 2$ can still perform exceptionally well in typical dynamical systems forecasting tasks (Gauthier et al., 2021; Shahi et al., 2022; Gauthier et al., 2022). However, applicability and performance of NVAR models beyond forecasting yet remain largely unexplored, particularly in the context of representing and comparing real-world time series data. In this regard, they present a promising solution to significantly enhance RC representations and kernels.

### 3.2 NVAR kernel

We present here the NVAR kernel (NVARk), which integrates the NVAR framework with RC-based kernel architectures. Given a pair of time series, $\mathbf{X}[i]$, $\mathbf{X}[j]$, with dimensionality $d_x$ of possibly different lengths, NVARk operates in three steps, which we present in the following paragraphs. Fig. 1 depicts a graphical reference, while Algorithm 1 explicits the algorithmic process.

---

[2] other forms of nonlinearity can be used depending on prior knowledge, but the polynomial functionals can be motivated as universal approximators of dynamical systems (Franz & Schölkopf, 2006)

---

**Algorithm 1** NVAR kernel

---

**input** MTS list $\mathbf{X}[1:N]$, number of lags $k$, lag size $s$, embedding size $\bar{d}_r$
1:  $\mathbf{L} = \{(a,p) \mid a \in \{1,...,d_x\}, p = \{1,...,k\}\}$ (dimension $a$ lagged $p$ times)
2:  $\mathbf{P} = \{((a,p),(b,q)) \mid a,b \in \{1,...,d_x\}, p,q = \{0,...,k\}\}$ (all possible products)
3:  $samples \leftarrow$ sample $(\bar{d}_r - d_x)$ elements from $\mathbf{L} \cup \mathbf{P}$ at random
4:  **for** $i$ in $[1:N]$ **do**
5:    $\mathbf{R}[i] \leftarrow \mathbf{X}[i]$
6:    **for** $elem$ in $samples$ **do**
7:      **if** $elem = (a,p)$ is in $\mathbf{L}$ **then**
8:        concatenate lag term: $\mathbf{R}[i] \leftarrow \left[\, \mathbf{R}[i] \,||\, \mathbf{x}^a_{t-p \times s}[i] \,\right]$
9:      **end if**
10:     **if** $elem = ((a,p),(b,q))$ is in $\mathbf{P}$ **then**
11:       concatenate nonlinear term: $\mathbf{R}[i] \leftarrow \left[\, \mathbf{R}[i] \,||\, \mathbf{x}^a_{t-p \times s}[i] * \mathbf{x}^b_{t-q \times s}[i] \,\right]$
12:     **end if**
13:   **end for**
14:   fit a linear model mapping $\mathbf{r}_{t-1}[i]$ to $\mathbf{r}_t[i]$
15:   vectorize fit parameters to obtain representation $\boldsymbol{\theta}[i]$
16: **end for**
17: for all representation pairs: $K_{ij} \leftarrow \text{RBF}(\boldsymbol{\theta}[i], \boldsymbol{\theta}[j])$
**output** K

---

**NVAR embeddings**   The general idea is to transform the time series using the NVAR map defined by Eq. 4, with $n = 2$. However, for a given choice of the number of lags $k$, the resulting number of additional dimensions scales with $O(d_x^2 k^2)$. For moderately high $d_x$, this can result in high collinearity and raise the curse of dimensionality in the following readout, making it slow and inaccurate. It also limits the possible choices of $k$ and, with it, the ability to capture longer-term memory effects. The method, as it is, then only seems tractable in the context of dynamical systems, where $d_x$ is usually very contained. To address this issue and make the method applicable to higher dimensional datasets, we propose to concatenate just a random subsample of all possible dimensions:

$$\mathbf{R}_{NVAR}|_{\bar{d}_r} = \mathbf{X} \,||\, \mathbf{r}_1 \,||\, \mathbf{r}_2 \,||\, ... \,||\, \mathbf{r}_{\bar{d}_r - d_x} \tag{5}$$

with $\{\mathbf{r}_a\}_{a=1}^{\bar{d}_r - d_x}$ randomly sampled columns in $[\mathbf{R}_{lag} \,||\, \mathbf{R}_{nonlin}]$ with uniform probability and $\bar{d}_r$ a threshold hyperparameter (notation $[i]$ is omitted for all terms and the same operations are applied to time series $[j]$). This converts the deterministic NVAR map into a random feature map. We support this choice in the following, by pointing out an unexplored connection of NVAR models to the theory of *state space reconstruction* (SSR) (Kantz & Schreiber, 2004; Strogatz, 2018).

It is generally true that we can interpret the observed data as a realization of more complex underlying dynamics ($\phi^*$), acting on some unknown states ($\mathbf{x}^*$): $\mathbf{x}^*_{t+1} = \phi^*(\mathbf{x}^*_t)$. SSR focuses on using the observed data ($\mathbf{x}$) to construct embedded states ($\mathbf{s}_t$) for which the dynamics is topologically equivalent to $\phi^*$, and thus a better representation of the underlying system (Casdagli et al., 1991). In this regard, the celebrated Takens' theorem (Takens, 1981) claims that a valid embedding, called *delay embedding* (Packard et al., 1980), can be formed by concatenating the input with its lags and, most importantly, guarantees that not all lags are needed, but only a *sufficient* number $\ell$: $\mathbf{s}_t = [x_t, x_{t-s}, x_{t-2s}, ..., x_{t-\ell s}]$. More recently, the work of Deyle & Sugihara (2011) presented a generalization of Takens' theorem to multivariate inputs in which, to match $\ell$, lags can be spread over different observed dimensions and alternative functions of the input can also be considered. All terms in the NVAR representation, i.e., lags and nonlinear terms, are reminiscent of this formulation, hence our choice of not considering all terms, but posing instead an upper threshold on the number of concatenation terms (Eq. 5).

This also allows for a different perspective on our approach. We can interpret $\mathbf{R}_{NVAR}|_{\bar{d}_r}$ as an attempt to create states for which the dynamics approximates the underlying one generating the data. As there is generally a loss of information in the generation process, a comparison based on such states would be more truthful than directly using the observed data. In practice, the presence of noise separates us from obtaining such exact states (Casdagli et al., 1991) and places our method under the field of *embedology* (Sauer et al., 1991), i.e., the building of delay-observation maps with special

features, overcoming the sensitivity to noise in the original Takens' formulation. This theoretical framework is more extensively discussed in Appendix A.

An alternative to Eq. 5 would have been to generate the full NVAR representation followed by a dimensionality reduction module as in Bianchi et al. (2020). In our preliminary evaluation, this proved to be computationally more expensive and led to poorer performance, possibly because of the dimensionality reduction not preserving the delay embedding structure.

**Linear readout**  Individual linear readouts (Eq. 3) are trained by ridge regression to solve a one-step-ahead prediction task on the same embedding states, i.e., using $\mathbf{y}_t = \mathbf{r}_{t+1}$. The parameters of each model are then extracted and vectorized to obtain the representations $\boldsymbol{\theta}[i]$ and $\boldsymbol{\theta}[j]$. This step removes the dependence on the length of the TS, which means that the kernel can compare TS of different lengths.

As the readout operates on the input concatenated with its lags and quadratic products, these representation vectors can be interpreted as encapsulating statistics related to the auto-mutual and mutual information within the dimensions of the input. In contrast, representations from RC methods encompass hardly interpretable reservoir dynamics.

**RBF aggregation**  Finally, a radial function is used to ensure the PSD property and form the kernel:

$$K(\mathbf{X}[i], \mathbf{X}[j]) = \exp\left\{ -\left|\left|\boldsymbol{\theta}[i] - \boldsymbol{\theta}[j]\right|\right|^2 / 2\gamma_{rbf}^2 \right\} \tag{6}$$

## 3.3  Hyperparameter setting

We display here the hyperparameters to construct NVARk and propose heuristic expressions.

**Number of lags** $(k)$ **and lag size** $(s)$  The parameters $k$ and $s$ play pivotal roles in the NVAR framework (Eq. 4) and, consequently, in our approach. These key parameters are of clear interpretation: $k$ controls the number of lags and extends memory capacity, i.e., capturing significant dependencies that extend further into the past; $s$ controls the spacing between lags, which manages the temporal resolution at which these dependencies are probed. In contrast, key RC parameters are arguably less interpretable.

In Takens' formulation, in the case of infinitely long, noise-free time series, there is no importance bias towards any specific lag $s$, and their number $k$ is the only relevant factor. This is often determined using the False Nearest Neighbours algorithm (Kennel et al., 1992; Wallot & Mønster, 2018). However, when dealing with limited noisy observations, the choice of $s$ becomes equally critical as it directly relates to the sampling frequency (Broomhead & King, 1986). In particular, using too high a value would force a modeling of uncorrelated points while too low would model noise. A correct choice of $s$ instead allows for extracting seasonality or other relevant trends as well as the underlying dynamics. In practice, for noisy systems, the choice of both $k$ and $s$ is inherently a difficult problem due to the absence of clear theoretical guidelines (Tran & Hasegawa, 2019). Despite that, interpretability can be exploited to propose task-dependent heuristics (Small & Tse, 2004), often requiring a trade-off between redundancy and the irrelevance of concatenated features.

In this work, we propose two settings. The first is a rule of thumb and proceeds as follows. We first apply a trend-filtering method to the time series, such as an $\ell_1$ trend (Kim et al., 2009) (this allows for a choice of $s$ that captures the temporal variability of the system rather than the superposed noise). We then set the product $k \cdot s$ as the average distance between peaks and hollows (dbp) (Kugiumtzis, 1996). Finally, we balance the contributions by adopting $s = k = \sqrt{\text{dbp}}$. Alternatively, in a supervised setting, both parameters are optimized by CV.

**Polynomial order** $(n)$  The polynomial order controls the amount of non-linearity in the embedding. As in most previous works, we found no need for further complexity than $n = 2$.

**Embedding dimension** $(\bar{d}_r)$  In Takens' formulation, an embedding size double the dimensionality of the underlying manifold is sufficient for reconstructing the system dynamics. However, when dealing with real-world data, this is unknown, and a trade-off must be struck between the omission of relevant features and redundancy, which can increase collinearity and computation times. In line with the scope of our work, we choose to give priority to a fair comparison with previous RC

methods (Bianchi et al., 2020), which identify $\bar{d}_r = 75$ as an optimal dimensionality before the readout. In Appendix C.1, we undertake a brief exploration to determine whether this value is also a reasonable choice for NVAR embeddings. While we did not observe any redundancy in univariate datasets, we did observe a performance drop after different $\bar{d}_r$ for some multivariate datasets. The study suggests $70 \lesssim \bar{d}_r \lesssim 100$ as a suitable range.

**Linear readout regularization** $(\lambda_{ridge})$  For the readout, we adopt the optimal regularization (OCReP) proposed in Cancelliere et al. (2015), i.e., set $\lambda_{ridge}$ to the product between the minimum and maximum singular value of $\mathbf{R}_{NVAR}|_{\bar{d}_r}$.

**RBF lengthscale** $(\gamma_{rbf})$  We set the lengthscale in Eq. 6 to the median pairwise distance between all the representations $\boldsymbol{\theta}[i]$ in the training set.

### 3.4 Asymptotic complexity

NVARk is based on the individual representations of time series. Hence, the computationally intensive step is performed only once per time series and results in $\mathcal{O}(N)$ complexity. This also implies that the computation can be parallelized along N. This provides a significant computational advantage over comparison-based kernels, which must perform expensive computations for all the relevant pairs of series, i.e., $\mathcal{O}(N^2)$.

In terms of scalability with the length of the series $(T)$, the advantage of NVAR over any recursive RC approach is the critical difference in how the hidden states are created. Eq. 5 is not iterative and only performs a fixed number $\bar{d}_r - d_x$ of concatenation operations, independent of the length of the time series. In contrast, reservoir-based methods are limited by the expensive recursive update of the reservoir state (Eq. 2), which is $\mathcal{O}(Td_r^2)$, where $d_r$ is the size of the reservoir (which is usually quite large). As evidence, improving the scalability of the reservoir update is an active area of research (Dong et al., 2020). The complexity of NVARk is limited to the only ridge regression of the output layer. This corresponds to solving a linear system with $T$ training examples and $D$ features, where D is at most $\bar{d}_r$. Such a system can usually be solved efficiently, e.g. by using LU (or Cholesky) factorization in asymptotically $\mathcal{O}(T\bar{d}_r^2)$.

As for the scalability with the dimensionality of the input series, all operations are bounded by $\bar{d}_r^2$, which we treat as a constant and do not discuss complexity in connection to this parameter. Overall, NVARk exhibits an asymptotic complexity at most $\mathcal{O}(NT\bar{d}_r^2)$.

## 4 Experimental evaluation

In this section, we present an experimental demonstration of the performance of NVARk. In line with the established literature (Cuturi, 2011; Cuturi & Doucet, 2011; Baydogan & Runger, 2016; Paparrizos et al., 2020), our main evaluation consists of time series classification. In particular, we pair different pre-computed kernel matrices with a Support Vector Machine (SVM) classifier (Steinwart & Christmann, 2008), a popular kernel classification method that lies between simple linear models and more advanced deep learning architectures. The evaluation is performed over 130 UTS datasets from the UCR archive and 19 MTS datasets from the UEA archive (Bagnall A. & E.) (Appendix B.1) and focuses on accuracy, execution time and scalability. In Appendix C.5, we present a snapshot of kernel Principal Component Analysis (kPCA) for dimensionality reduction and visualization.

Performance of the proposed approach is investigated under two different hyperparameter settings. We indicate with NVARk the setting in which $k$ and $s$ are determined by the general proposed rule of thumb of Sec. 3.3. The maximum number of dimensions is set to $\bar{d}_r = 75$. As for the other hyperparameters, we follow the setting given in Sec. 3.3. We instead indicate with NVARk* the configuration in which $k$ and $s$ are optimized in a CV loop over a small grid. As the datasets cover a wide range of series lengths, the CV grid is slightly adapted consequently (Appendix. B.2).

The comparison is performed against previously proposed kernels: the **reservoir-model Echo State Network** (rmESN) (Bianchi et al., 2020) as the reservoir equivalent to our approach. We set $D = 75$ as the dimensionality before the linear readout. This allows us to directly compare the quality of the extracted dynamic features with or without a reservoir; the **Time Cluster Kernel** (TCK) (Mikalsen et al., 2018) as representative of model-based kernels and SOTA for MTS; the **Shift-INvariant**

**Kernel** (SINK) (Paparrizos & Franklin, 2019) as representative of Fourier-based methods and SOTA for UTS. A comparison with the Global Alignment Kernel (GAK) (Cuturi, 2011) is presented in Appendix C.4. Guidelines in Appendix B.2 contain the hyperparameter settings and links to the code implementations for NVARk and all the aforementioned baselines.

Despite the fact that some datasets in our benchmark can be also investigated using Deep Learning (DL) methods (Ismail Fawaz et al., 2019), the use case for kernel methods is substantially different, usually focused on small datasets, where the application of DL is not trivial. In particular, kernels can compute the similarity between just two time series right away. Consequently, in the literature, kernels are usually not directly tested against DL. Nevertheless, in our work, we compare to ESN-based representations, which are known to be competitive with a variety of DL architectures (Bianchi et al., 2020; Shahi et al., 2022). Thus, ESN can be seen as a proxy for more complex architectures.

As preprocessing, we apply zero-padding to match all series to the maximum length in each dataset (to allow comparison with TCK) and follow the common practice of standardizing all series to zero mean and unit variance. For each dataset-method pair, we allow for a maximum execution time of 20 hours.

## 4.1 SVM classification of UTS

We show here SVM classification for univariate datasets. SVM requires the setting of the hyperplane regularization $C$, which we optimize in a CV loop from logarithmically spaced values in $[10^{-3}, 10^3]$. This is the only free parameter in all methods except NVARk*, in which it is jointly optimized with $k$ and $s$. For the CV loop, we adopt a 10-fold CV with the size of the validation set corresponding to 33% of the train set. SVM accuracy for each dataset is obtained by averaging 10 repetitions with different seeds. Tab. 1 reports the average classification accuracy for all kernels within the presented framework, as well as the relative ranking (individual results are reported in Appendix D). Overall,

Table 1: SVM classification of UTS datasets, performance metrics across 130 datasets.

|  | NVARk* | NVARk | rmESN | TCK | SINK |
|---|---|---|---|---|---|
| AVERAGE SCORE | 0.803 | **0.779** | 0.757 | 0.726 | 0.771 |
| AVERAGE RANK | / | **2.14** | 2.62 | 2.85 | 2.20 |
| N FIRST RANKED | / | **52** | 26 | 23 | 43 |
| MEDIAN TIME (S) / DATASET | 14.27 | **0.59** | 40.87 | 44.24 | 23.73 |

NVARk obtains very competitive results, with the highest average accuracy and rank. The case of univariate datasets is a comfortable scenario for NVARk to operate in as, in most cases, all possible terms in $\mathbf{R}_{NVAR}$ can be considered without exceeding $\bar{d}_r$ and the effect of collinearity is minimized. NVARk and SINK appear indistinguishable under a Friedman + Nemenyi statistical test, which places NVARk under the SOTA in terms of accuracy. Interestingly, the difference between NVARk and rmESN is significant at the 95% level with p-value$_{NVARk-rmESN} = 0.013$. Results for NVARk* show that the performance of NVARk can be very efficiently improved by a CV optimization of just the two integer embedding hyperparameters.

## 4.2 Execution time and scalability

In terms of execution time, the scalability properties of NVARk (Sec. 3.4) manifest in exceptional performance. In Fig. 2 (left) and Tab. 1 (last row), we compare running times to compute the train-train and the train-test kernel matrices and observe that the improvement of NVARk is consistent across datasets and baselines, with speedup factors up to $\times 10^3$ in the high-$T$ and high-$N$ regimes.

Results for NVARk* can be achieved with a slowdown factor of $\sim 25$ (parallelized grid search + 1 iteration with optimal parameters) with respect to the heuristic-based NVARk. Interestingly, for 113/130 datasets, NVARk* execution time is faster than a single iteration for rmESN.

For a study of the scaling, in Fig. 2 (middle, right) we progressively vary the length of the *Abnormal-Heartbeat* dataset (by interpolation) and the number of training samples (by random sampling) in the *Crop* dataset. Results show that NVARk performs exceptionally well in all regimes, while rmESN

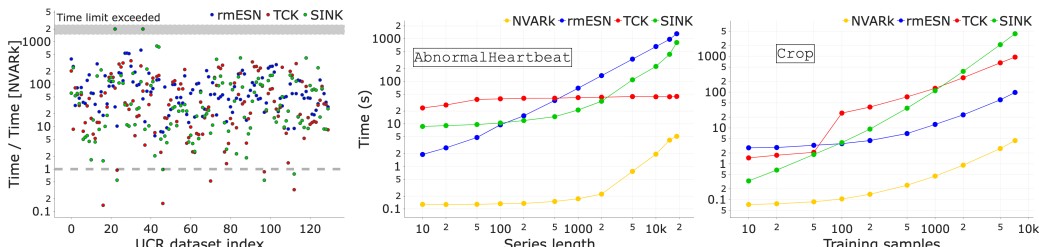

Figure 2: **Left:** For all univariate datasets, the plot shows the ratio between the execution time to compute the train-train and train-test kernels of different methods and NVARk. **Middle**: Scalability of kernels with the series length. **Right**: Scalability of kernels with the training size.

scales poorly with the length of the series and TCK and SINK with the size of the training sample. NVARk, instead, is advantaged by its non-recursive structure and linear complexity in $N$.

### 4.3 SVM classification of MTS

We present here SVM classification over the 19 MTS datasets. We do not include SINK as there is no extension or evaluation provided for MTS. Aside from that, the evaluation is conducted identically to the univariate case. For multivariate datasets, we expect to drift away from the ideal operating

Table 2: SVM classification of MTS datasets. Average across 10 seeds. Best accuracy is in bold.

| DATASET | NVARK* | NVARK | RMESN | TCK |
|---|---|---|---|---|
| SPOKENARABICDIG. | $0.980_{\pm 0.004}$ | $0.980_{\pm 0.003}$ | $0.948_{\pm 0.004}$ | $\mathbf{0.985}_{\pm 0.002}$ |
| JAPANESEVOWELS | $0.981_{\pm 0.008}$ | $\mathbf{0.975}_{\pm 0.007}$ | $0.968_{\pm 0.004}$ | $0.961_{\pm 0.004}$ |
| PENDIGITS | $0.986_{\pm 0.000}$ | $\mathbf{0.983}_{\pm 0.000}$ | $0.976_{\pm 0.001}$ | $0.954_{\pm 0.001}$ |
| RACKETSPORTS | $0.889_{\pm 0.019}$ | $0.847_{\pm 0.007}$ | $0.859_{\pm 0.008}$ | $\mathbf{0.872}_{\pm 0.006}$ |
| LSST | $0.566_{\pm 0.009}$ | $0.543_{\pm 0.006}$ | $\mathbf{0.597}_{\pm 0.009}$ | $0.474_{\pm 0.008}$ |
| LIBRAS | $0.972_{\pm 0.000}$ | $\mathbf{0.900}_{\pm 0.000}$ | $0.824_{\pm 0.010}$ | $0.793_{\pm 0.010}$ |
| FINGERMOVEMENTS | $0.572_{\pm 0.034}$ | $\mathbf{0.581}_{\pm 0.032}$ | $0.576_{\pm 0.022}$ | $0.505_{\pm 0.030}$ |
| NATOPS | $0.896_{\pm 0.020}$ | $\mathbf{0.895}_{\pm 0.019}$ | $0.828_{\pm 0.015}$ | $0.806_{\pm 0.016}$ |
| CHARACTERTRAJ. | $0.982_{\pm 0.000}$ | $0.915_{\pm 0.008}$ | $\mathbf{0.987}_{\pm 0.002}$ | $0.974_{\pm 0.002}$ |
| ERING | $0.898_{\pm 0.021}$ | $0.864_{\pm 0.025}$ | $0.814_{\pm 0.014}$ | $\mathbf{0.957}_{\pm 0.005}$ |
| BASICMOTIONS | $1.000_{\pm 0.000}$ | $0.938_{\pm 0.027}$ | $0.995_{\pm 0.011}$ | $\mathbf{1.000}_{\pm 0.000}$ |
| ARTICULARYWORDREC. | $0.978_{\pm 0.008}$ | $0.976_{\pm 0.005}$ | $\mathbf{0.986}_{\pm 0.006}$ | $0.983_{\pm 0.003}$ |
| EPILEPSY | $0.986_{\pm 0.000}$ | $\mathbf{0.986}_{\pm 0.000}$ | $0.971_{\pm 0.005}$ | $0.947_{\pm 0.005}$ |
| UWAVEGESTURELIB. | $0.946_{\pm 0.003}$ | $\mathbf{0.939}_{\pm 0.011}$ | $0.862_{\pm 0.007}$ | $0.868_{\pm 0.007}$ |
| SELFREGULATIONSCP1 | $0.746_{\pm 0.017}$ | $0.688_{\pm 0.032}$ | $0.769_{\pm 0.019}$ | $\mathbf{0.827}_{\pm 0.007}$ |
| SELFREGULATIONSCP2 | $0.566_{\pm 0.017}$ | $\mathbf{0.567}_{\pm 0.028}$ | $0.511_{\pm 0.027}$ | $0.488_{\pm 0.014}$ |
| CRICKET | $0.979_{\pm 0.010}$ | $\mathbf{0.986}_{\pm 0.000}$ | $0.982_{\pm 0.006}$ | $0.958_{\pm 0.000}$ |
| STANDWALKJUMP | $0.433_{\pm 0.047}$ | $\mathbf{0.607}_{\pm 0.049}$ | $0.373_{\pm 0.056}$ | $0.433_{\pm 0.079}$ |
| EIGENWORMS | $0.944_{\pm 0.009}$ | $\mathbf{0.950}_{\pm 0.012}$ | $0.939_{\pm 0.013}$ | $0.594_{\pm 0.000}$ |
| AVERAGE SCORE | 0.858 | **0.848** | 0.830 | 0.809 |
| AVERAGE RANK | / | **1.68** | 2.05 | 2.26 |
| N FIRST RANKED | / | **11** | 3 | 5 |
| MEDIAN TIME (S) / DATASET | 19.67 | **3.87** | 26.47 | 184.08 |

scenario of NVARk. In particular, all the possible dynamic features may not be equally effective and, by randomly selecting a few, important correlations may be missed or spurious ones captured. Despite that, the experiments in Tab. 2 show surprisingly good results, even for datasets for which $\bar{d}_r$ is just a fraction of all possible concatenation terms. We believe that this finding strengthens the connection to the generalized Takens' theorem. It is also interesting to notice good performance in diverse cases of time series that do not immediately arise from dynamical systems. Exploring this aspect constitutes interesting future works, e.g., time series representation learning and state space reconstruction as a possible underpinning.

Finally, we observe that the two hyperparameter configurations, NVARk* and NVARk, tend to be ranked next to each other. This supports the quality of the proposed general setting, which allows the kernel to extract meaningful features while preserving great computational efficiency.

### 4.4 Ablation study

In order to tease out which NVARk component is most important, we here present an ablation study. We separately consider fitting the linear readout directly on the input time series (no concatenation), sampling only linear terms from Eq. 5, and sampling only non-linear terms. This is an interesting ablation study that, to the best of our knowledge, has not been investigated in previous works. Results are presented in Tab. 3.

Table 3: Ablation study across 130 UCR univariate and 19 multivariate UEA datasets

|  |  | no concat | linear only | non-linear only | all | balanced |
|---|---|---|---|---|---|---|
| SVM score for | mean | 0.536 | 0.707 | 0.764 | 0.779 | 0.779 |
| univariate datasets | std | 0.010 | 0.008 | 0.006 | 0.006 | 0.006 |
| SVM score for | mean | 0.765 | 0.840 | 0.840 | 0.848 | 0.849 |
| multivariate datasets | std | 0.012 | 0.008 | 0.013 | 0.014 | 0.012 |

As a general trend, we observe that using all terms leads to the best results, followed by *non-linear only*, *linear only*, and fitting directly on the input series. Note also that results are in line with our interpretation using the generalized Taken's theorem. For univariate datasets, *linear only* tends to underperform, as the total number of concatenated dimensions is very small (equal to $k$) and may be not sufficient in reconstructing the underlying manifold. In the multivariate case, the two variants perform very similarly, as the lags of all attributes may already constitute a considerable pool of terms.

To conclude, we consider an additional variant of Eq. 5 in which we first concatenate all the lag terms, and then fill the remaining dimensions, up to $\bar{d}_r$, by sampling nonlinear terms. Interestingly, for multivariate datasets, this leads to slightly better performance and a reduction in variance. We believe this might eventually compensate for the imbalance between all possible linear and non-linear terms, while also reducing the chance of adding noise from uncorrelated dimensions.

## 5 Conclusions

We have proposed a kernel for time series that integrates the NVAR framework into reservoir-based kernel architectures. NVARk compares time series based on the linear dynamics of NVAR embeddings, which are built from concatenating lags and nonlinear functionals to the original series. In terms of accuracy, NVARk outperforms the corresponding RC architecture. Computationally, it is exceptionally efficient and based on a few integer hyperparameters, which together allow for further improvement of the results with simple supervised grid-based optimization.

As for future work directions, we believe it would be effective to target specific weaknesses of NVARk. In particular, as the input dimensionality of the time series approaches the maximum embedding dimension ($\bar{d}_r$), the proposed approach finds little margin for concatenating more lags and non-linear terms. Such cases would require substantially increasing $\bar{d}_r$ and, consequently, place excessive strain on the linear readout, which inherently possesses limited expressive capacity. We then expect the effectiveness of NVARk to decrease. To overcome this, one can consider replacing the random sampling of dimensions with more refined strategies that prioritize the selection of meaningful terms. Alternatively, it would be interesting to explore how different linear layers, e.g. Lasso, would perform in this regime. Alongside these, different unsupervised strategies can be considered to learn optimal values for the embedding parameters. In particular, we note an unexplored affinity with *signature transforms* (Chevyrev & Kormilitzin, 2016), which we plan on deepening to infer optimal settings for the maximum dimensionality of the NVAR embedding.

## Acknowledgements

G.D. acknowledges the Beckers Group for funding this research. V.V.G. thanks Leverhulme Trust for support via the Leverhulme Research Centre for Functional Materials Design. We would like to thank Chao Huang, Filippo Maria Bianchi, Davide Bacciu, and Claudio Gallicchio for their fruitful feedback and support for the work. Also, we would like to thank all the reviewers for their invaluable comments.

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

# Appendix

## A   Takens' delay embeddings

This section introduces the field of *state space reconstruction* in dynamical system theory (Strogatz, 2018) and deepens the theoretical interpretation of Sec. 3.2. As in Deyle & Sugihara (2011), we start by defining some key concepts that might be useful to introduce the reader to the subject:

- a *manifold* is, intuitively, a generalized surface $M \subseteq \mathbb{R}^N$ that locally looks like an Euclidean space. More rigorously, locally diffeomorphic to some $\mathbb{R}^k$;
- a *dynamical system* $\phi : M \rightarrow M$ is a diffeomorphism defining, for discrete times, trajectories on a manifold M;
- an *embedding* is a 1:1 smooth map $f : M_1 \rightarrow M_2$ between two manifolds. Embeddings preserve topological properties, e.g. resolves all trajectories without crossings, and geometrical invariants, such as eigenvalues and fixed points;
- an *observation function* $y : M \rightarrow \mathbb{R}^D$ is any function that assigns a vector to each point on M.

It is generally true that we can imagine an observed MTS $\mathbf{X} \in \mathbb{R}^{T \times D}$ as a realization of a more complex and unknown smooth dynamical system $\phi^*$, acting on states $\mathbf{x}^*$ on an unknown manifold $M^*$:

$$\mathbf{x}_{t+1}^* = \phi^*(\mathbf{x}_t^*) \tag{7}$$

The observed and original values are related by an observation function $y^* : M^* \rightarrow \mathbb{R}^D$ that projects the underlying states onto what we observe:

$$\mathbf{x}_t = y^*(\mathbf{x}_t^*) \quad (+ \boldsymbol{\epsilon}_t) \tag{8}$$

This projection can drastically reduce the available information, but one can wonder to which extent it is possible to recover the original system when only the observed vectors $(\mathbf{x}_t)$ are accessible. We focus here on the *state reconstruction problem*, which is that of finding a valid embedding $f : M^* \rightarrow \mathbb{R}^n$ of the original manifold $M^*$:

$$\mathbf{s}_t = f(\mathbf{x}_t^*) \tag{9}$$

This is, by using the observed states $(\mathbf{x}_t)$, creating richer states $(\mathbf{s}_t)$ for which the dynamics is topologically equivalent to the unknown $\phi^*$. In fact, if $f$ is an embedding, then smooth dynamics $\Phi$ is induced on the space of reconstructed vectors:

$$\Phi(\mathbf{s}_t) = f \circ \phi^* \circ f^{-1}(\mathbf{s}_t) \tag{10}$$

and is equivalent to $\phi^*$. If we succeed, instead of the observed $\mathbf{x}$, we can use the reconstructed vectors $\mathbf{s}$ as a better representation of the underlying evolving system $\mathbf{x}^*$.

Whitney's theorem (Whitney, 1936) states the general result that every compact m-dimensional manifold admits an embedding in $\mathbb{R}^{2m+1}$. For dynamical systems this means that, a set of $2m + 1$ observation functions, where $y_k : M \rightarrow \mathbb{R}$, is sufficient to construct a valid embedding:

$$\mathbf{s}_t = [y_1(\mathbf{x}_t^*), y_2(\mathbf{x}_t^*), ..., y_{2m+1}(\mathbf{x}_t^*)] \tag{11}$$

In the case of scalar observations (D=1), Takens' theorem (Takens, 1981) considers a single observation function $(y)$, i.e., the projection on the only observed component, but acting on multiple lagged versions of the system:

$$\begin{aligned} \mathbf{s}_t &= [y(\mathbf{x}_t^*), y(\phi(\mathbf{x}_t^*)), y(\phi^2(\mathbf{x}_t^*)), ..., y(\phi^{2m}(\mathbf{x}_t^*))] \\ &= [x_t, x_{t-\tau}, ..., x_{t-2m\tau}] \end{aligned} \tag{12}$$

The idea behind Eq. 12 is that the information about the unobserved state variables is contained in the relations between past and future values of the observed time series and a $2m + 1$ dimensional embedding (often less) is sufficient to extract that.

Generalized versions of Takens' theorem have also been formulated. In particular, we find in Deyle & Sugihara (2011) an extension to cases where lags of multiple observation functions can be used to create the embedding. This is a useful result in cases where one cannot consider many lags, but can instead make use of fewer lags spread across multiple available dimensions. In more detail, the allowed set of observation functions is any $\{y_k\}_{k=1}^{2m+1}$ containing terms $\{y_r\}$ that cannot be written as a lagged version of any other element in the set and terms $\{y_q\}$ that satisfy $y_q = y_r(\phi^b(\mathbf{x}^*))$ for a choice of $b$. Both Takens' and Whitney's theorems can be recovered from this formulation: it reduces to the case of Takens' when only one $\{y_r\} = y$ is chosen and all the others belong to the family of $\{y_q\}$; it reduces to the case of Whitney's when $\{y_r\} = \{y_k\}$ and $\{y_q\} = \emptyset$.

All terms in the NVAR representation are reminiscent of this formulation. This points to the direction of $\mathbf{R}_{NVAR}$ being a closer approximation of the underlying state than the initial data $\mathbf{X}$ and also suggests that the final number of dimensions plays a more important role than which are included and which are not. In practice, the dimensionality $m$ of the original manifold $M^*$ is unknown, making the selection of the embedding dimensionality a potentially rich field of study. On top of that, the observed data may be corrupted by multiple noise sources, which increases the importance of sampling the series correctly with a meticulous choice of the lag size $(\tau \equiv s)$.

# B  Experimental evaluation details

## B.1  Datasets

Details of the 130 Univariate and 19 Multivariate datasets are reported in Tab. 4. Both cover an extensive range of data points and series lengths. From the UEA archive, we have excluded 6 high-dimensional datasets for which the application of the NVAR kernel is not appropriate, and 5 datasets for which SVM accuracy was $< 0.5$ for all approaches, which indicates that kernel methods are, in general, not a suitable solution.

Table 4: Attributes of UTS and MTS datasets. The splitting between the training and testing set is provided by the respective archive.

| UTS DATASET | $\mathbf{N}_{train}$ | $\mathbf{N}_{test}$ | $\mathbf{T}$ | | $\mathbf{D}$ | $\mathbf{N}_{class}$ |
|---|---|---|---|---|---|---|
| 130 DATASETS | $16 \div 8926$ | $20 \div 16800$ | $15 \div 18530$ | | 1 | $2 \div 60$ |

| MTS DATASET | $\mathbf{N}_{train}$ | $\mathbf{N}_{test}$ | $\mathbf{T}_{min}$ | $\mathbf{T}_{max}$ | $\mathbf{D}$ | $\mathbf{N}_{class}$ |
|---|---|---|---|---|---|---|
| SPOKENARABICDIG. | 6599 | 2199 | 4 | 93 | 13 | 10 |
| JAPANESEVOWELS | 270 | 370 | 7 | 29 | 12 | 9 |
| PENDIGITS | 7494 | 3498 | 8 | 8 | 2 | 10 |
| RACKETSPORTS | 151 | 152 | 30 | 30 | 6 | 4 |
| LSST | 2459 | 2466 | 36 | 36 | 6 | 14 |
| LIBRAS | 180 | 180 | 45 | 45 | 2 | 15 |
| FINGERMOV. | 316 | 100 | 50 | 50 | 28 | 2 |
| NATOPS | 180 | 180 | 51 | 51 | 24 | 6 |
| CHARACTERTRAJ. | 1422 | 1436 | 60 | 182 | 3 | 20 |
| ERING | 30 | 270 | 65 | 65 | 4 | 6 |
| BASICMOTIONS | 40 | 40 | 100 | 100 | 6 | 4 |
| ARTICULARYWORDREC. | 275 | 300 | 144 | 144 | 9 | 25 |
| EPILEPSY | 137 | 138 | 206 | 206 | 3 | 4 |
| UWAVEGESTURELIB. | 120 | 320 | 315 | 315 | 3 | 8 |
| SELFREGULATIONSCP1 | 268 | 293 | 896 | 896 | 6 | 2 |
| SELFREGULATIONSCP2 | 200 | 180 | 1152 | 1152 | 7 | 2 |
| CRICKET | 108 | 72 | 1197 | 1197 | 6 | 12 |
| STANDWALKJUMP | 12 | 15 | 2500 | 2500 | 4 | 3 |
| EIGENWORMS | 128 | 131 | 17984 | 17984 | 6 | 5 |

## B.2  Implementations and details for reproducibility

For reproducibility, we list here the parameter setting and the code implementation for each method. Experiments were run on a 32-core AMD Ryzen™ Threadripper PRO CPU.

**NVAR kernel (NVARk)**

- the allowed number of lags $k = \sqrt{\text{average distance between peaks and hollows}}$, after series have been $\ell_1$ trend-filtered (Kim et al., 2009);

- lag size $s = k$;

- order of the polynomial functionals $n = 2$;

- maximum dimensionality of the embedding $\bar{d}_r = 75$;

- readout regularization set by the OCReP method (Cancelliere et al., 2015);

- RBF lengthscale as the median pairwise distance between the representations.

An easy-to-use Python implementation of the NVAR kernel is made publicly available at `https://github.com/gdefe/nvark-kernel`.

Table 5: Grid values for the CV optimization of the NVARk embedding parameters.

| DATASET | CV GRID |
|---|---|
| $T < 400$ | K=[1,2,3,4,5,K*], S=[1,2,3,4,5,S*] |
| $T \geq 400$ | K=[1,2,3,4,5,10,20,K*], S=[1,5,10,20,S*] |
| SPOKENARABICDIGITS | K=[1,2,3,4,K*], S=[1,2,3,4,S*] |
| JAPANESEVOWELS | " |
| PENDIGITS | " |
| RACKETSPORTS | " |
| LSST | " |
| LIBRAS | " |
| FINGERMOVEMENTS | " |
| NATOPS | " |
| CHARACTERTRAJECTORIES | K=[1,2,3,4,10,20,K*], S=[1,5,20,S*] |
| ERING | " |
| BASICMOTIONS | " |
| ARTICULARYWORDRECOGNITION | " |
| EPILEPSY | " |
| UWAVEGESTURELIBRARY | " |
| SELFREGULATIONSCP1 | " |
| SELFREGULATIONSCP2 | " |
| CRICKET | " |
| STANDWALKJUMP | " |
| EIGENWORMS | " |

**optimized NVAR kernel (NVARk\*)**   The lag size $s$ and the allowed number of lags $k$ are optimized in a CV loop scrolling a small grid. Values for the grid are reported in Tab. 5. $k^*$ and $s^*$ refer to the values obtained from the rule of thumb.

In general, one should allow for larger values for $k$ and $s$ in the case of longer series. A higher $k$ allows capturing longer-term dynamics, while a larger $s$ avoid oversampling the series. In the case of multivariate datasets, we observed that many lags are hardly required. This is possible because few lags spread across different dimensions can be sufficient to construct the embedding. Also, the pool of features ($[\mathbf{R}_{lag} \| \mathbf{R}_{nonlin}]$) largely increases with the dimensionality of the dataset and $k$, which affects the stability of the approach.

**reservoir-model ESN (rmESN)**   For the rmESN, we follow the setting given in Bianchi et al. (2020):

- number of internal reservoir units $R = 800$;
- spectral radius $\rho = 0.99$;
- no leakage;
- percentage of non-zero connectivity within the reservoir states $\beta = 0.25$;
- input scaling $\omega = 0.15$;
- noise level in the reservoir update $\epsilon_t = 0.001$;
- transient dropped states $n_{drop} = 5$;
- number of dimensions after the dimensionality reduction module $D = 75$ (via tenor-PCA, also proposed in the same paper);
- favoring comparability with NVARk, we set the readout regularization by the OCReP method;
- favoring comparability with NVARk, we set the RBF lengthscale to the median pairwise distance between the representations.

As implementation, we used the public Python code provided by the authors: `https://github.com/FilippoMB/Time-series-classification-and-clustering-with-Reservoir-Computing`.

**Time Cluster Kernel (TCK)**  For TCK, we follow the guidelines given in the original paper and in the provided code. For a dataset of $N$ samples and $D$ attributes, these are:

- minimum percentage of subsample $minN = 0.8$;
- min number of attributes for each GMM $minD = 1$ for univariate datasets, $minD = 2$ for multivariate datasets;
- max number of attributes for each GMM $maxD = min(ceil(0.9 * D), 15)$;
- min length of time segments for each GMM $minT = 6$;
- max length of time segments for each GMM $maxT = min(floor(0.8 * T), 25)$;
- max number of mixture components for each GMM $C = 10$ if $N < 100$, $C = 40$ otherwise;
- number of randomizations for each number of components $G = 30$;
- number of iterations $I = 20$.

The Matlab implementation of TCK that we use is part of the code available at `https://github.com/FilippoMB/TCK_AE`.

**Global Alignment Kernel (GAK)**  We set the multiplicative factor for the bandwidth of the radial function to $\sigma_{GAK} = 2$. Python code is available from the *tslearn* package (Tavenard et al., 2020). As reported in the package documentation, the method can lead to numerical issues for long series.

**Shift INvariant Kernel (SINK)**  For SINK, we adopted the Python implementation provided by the author at `https://github.com/TheDatumOrg/grail-python`. We use $\gamma = 5$ and retain all the energy after Fourier-transforming the series.

## C  Additional experiments

### C.1  Feature redundancy

For studying whether previously adopted values for the embedding dimensionality ($\bar{d}_r$) are also suitable for NVARk, we adopted an experimental approach and studied how many features can be included before experiencing redundancy. We do so by plotting, for a few univariate and multivariate datasets, the accuracy of an SVM classifier against the $\bar{d}_r$ parameter (Fig. 3). While do not observe redundancy in univariate datasets,

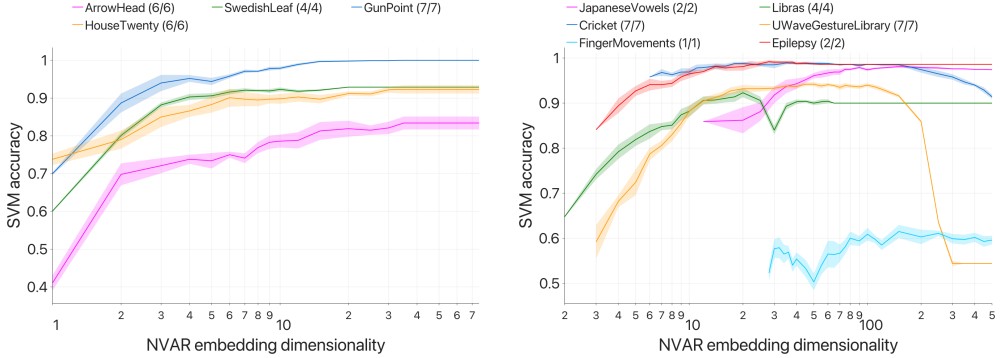

Figure 3: For some univariate (left) and multivariate (right) time series datasets, the plots show the accuracy of an SVM classifier against the NVAR embedding dimensionality $\bar{d}_r$. Embedding parameters are indicated as $(k/s)$ and determined via the proposed rule of thumb. Features redundancy manifests for some multivariate datasets (*Cricket*, *UWaveGestureLibrary*) as an accuracy drop.

we do observe a drop in performance after different $\bar{d}_r$ for some multivariate datasets. We identify the region $70 \leq \bar{d}_r \leq 100$ as a good, though non-optimal, setting, which is in line with previous RC studies (Bianchi et al., 2020).

## C.2 Sensitivity analysis

In this section, we address the sensitivity of NVARk to the different hyperparameters:

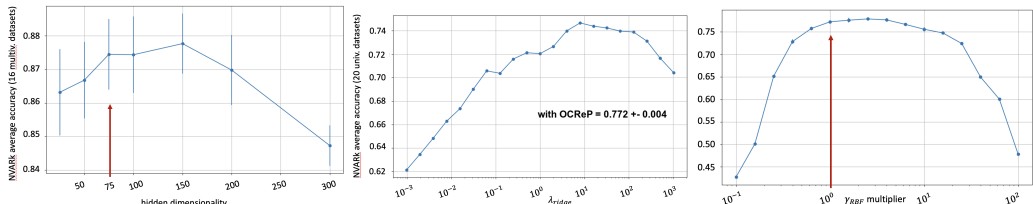

Figure 4: Sensitivity of NVARk to the hidden dimensionality (left), evaluated across 16 multivariate datasets; sensitivity to the regularization parameter (center) and RBF lengthscale (right), evaluated across 20 randomly sampled univariate datasets. Red arrows indicate our parameter's choice.

- $\bar{d}_r$: as we show in Fig. 4 (left), the average performance for multivariate datasets is not considerably impacted in the range $\bar{d}_r \in [70, 150]$. As expected, we observe a significant drop in performance for high values (redundancy).

- $\lambda_{\text{ridge}}$: we show in Fig. 4 (center) little sensitivity in the region $\lambda_{ridge} \in [5, 50]$. Results are averaged across a pool of 20 randomly sampled univariate datasets. The employed OCReP method (Cancelliere et al., 2015) outperforms the best choice by $\sim 2\%$, as it adapts the value to each individual dataset.

- $\gamma_{\text{rbf}}$: in Fig. 4 (right), we range different multiplicative factors for the median pairwise distance between all representations and show little sensitivity in a neighborhood of our choice $\gamma_{\text{rbf}_{\text{mul}}} = 1$.

- $k$ and $s$: The sensitivity to the embedding parameters strongly depends on the dataset, although it generally tends to be high. In response, we introduce our dataset-specific heuristic adapting ideas from a precedent study (Kugiumtzis, 1996), which we find works well in practice. A reasonable idea of the average sensitivity to these parameters can be obtained by comparing the performance of NVAR, where $k$ and $s$ are chosen by the heuristic, to the performance of NVARk*, where $k$ and $s$ are optimized. Unfortunately, understanding sensitivity by considering other settings is challenging, as the existing literature predominantly focuses on the noise-free case. Furthermore, there is no strong evidence for the wider applicability of such approaches outside dynamical systems.

## C.3 Uneven lengths

NVARk can process and compare time series of different lengths. We report in Tab. 6 the SVM accuracy scores for three datasets with uneven lengths.

Table 6: SVM classification accuracy for datasets with uneven lengths marked with the [dL] notation. For comparison, here are copied the respective results from Sec. 4.3 after zero padding preprocessing.

| DATASET | NVARK* | NVARK |
|---|---|---|
| SPOKENARABICDIGITS [DL] | $0.971_{\pm 0.003}$ | $0.971_{\pm 0.003}$ |
| SPOKENARABICDIGITS | $0.980_{\pm 0.004}$ | $0.980_{\pm 0.003}$ |
| JAPANESEVOWELS [DL] | $0.973_{\pm 0.007}$ | $0.975_{\pm 0.004}$ |
| JAPANESEVOWELS | $0.981_{\pm 0.008}$ | $0.975_{\pm 0.007}$ |
| CHARACTERTRAJECTORIES [DL] | $0.983_{\pm 0.005}$ | $0.933_{\pm 0.000}$ |
| CHARACTERTRAJECTORIES | $0.982_{\pm 0.000}$ | $0.915_{\pm 0.008}$ |

## C.4 Comparison with Global Alignment Kernel

We compare here the performance of the NVAR kernel against the popular Global Alignment Kernel (GAK) baseline from Cuturi (2011). This has been moved from the main text as authors of SINK have already demonstrated superior performance to GAK, following a similar evaluation. As reported in the *tslearn* package documentation, the provided implementation can lead to numerical issues when used with relatively long time series. To face this, we evaluate GAK on time series linearly interpolated to $T = 400$ when their length exceeds this threshold. We report in Tab. 7 the average results across 122 univariate datasets. We report in Tab. 8 results for the multivariate datasets. Datasets interpolated to T=400, before being processed by GAK, are marked with the symbol (*).

Table 7: Average performances for NVARk and GAK across 130 univariate datasets.

| | NVARk* | NVARk | GAK |
|---|---|---|---|
| AVERAGE SCORE | 0.803 | **0.779** | 0.728 |
| N FIRST RANKED | / | **77** | 56 |

Table 8: SVM classification of MTS datasets. Average across 10 seeds. Best accuracy is in bold.

| DATASET | NVARk | GAK |
|---|---|---|
| SPOKENARABICDIG. | $\mathbf{0.980}_{\pm 0.003}$ | *time limit* |
| JAPANESEVOWELS | $\mathbf{0.975}_{\pm 0.007}$ | $0.974_{\pm 0.002}$ |
| PENDIGITS | $\mathbf{0.983}_{\pm 0.000}$ | $0.978_{\pm 0.000}$ |
| RACKETSPORTS | $\mathbf{0.847}_{\pm 0.007}$ | $0.836_{\pm 0.008}$ |
| LSST | $\mathbf{0.543}_{\pm 0.006}$ | $0.500_{\pm 0.000}$ |
| LIBRAS | $\mathbf{0.900}_{\pm 0.000}$ | $0.783_{\pm 0.007}$ |
| FINGERMOVEMENTS | $\mathbf{0.581}_{\pm 0.032}$ | $0.565_{\pm 0.035}$ |
| NATOPS | $0.895_{\pm 0.019}$ | $\mathbf{0.918}_{\pm 0.003}$ |
| CHARACTERTRAJ. | $0.915_{\pm 0.008}$ | $\mathbf{0.988}_{\pm 0.000}$ |
| ERING | $0.864_{\pm 0.025}$ | $\mathbf{0.931}_{\pm 0.002}$ |
| BASICMOTIONS | $0.938_{\pm 0.027}$ | $\mathbf{0.995}_{\pm 0.011}$ |
| ARTICULARYWORDREC. | $0.976_{\pm 0.005}$ | $\mathbf{0.980}_{\pm 0.000}$ |
| EPILEPSY | $\mathbf{0.986}_{\pm 0.000}$ | $0.845_{\pm 0.004}$ |
| UWAVEGESTURELIB. | $\mathbf{0.939}_{\pm 0.011}$ | $0.865_{\pm 0.002}$ |
| SELFREGULATIONSCP1 | $0.688_{\pm 0.032}$ | $\mathbf{0.902}_{\pm 0.008}(*)$ |
| SELFREGULATIONSCP2 | $\mathbf{0.567}_{\pm 0.028}$ | $0.526_{\pm 0.017}(*)$ |
| CRICKET | $\mathbf{0.986}_{\pm 0.000}$ | $0.958_{\pm 0.000}(*)$ |
| STANDWALKJUMP | $\mathbf{0.607}_{\pm 0.049}$ | $0.333_{\pm 0.172}(*)$ |
| EIGENWORMS | $\mathbf{0.950}_{\pm 0.012}$ | $0.482_{\pm 0.005}(*)$ |
| AVERAGE SCORE | **0.848** | 0.756 |
| N FIRST RANKED | **13** | 6 |

## C.5 Example kPCA visualization

Despite the underlying equivalence between NVAR and simple RC, better performance can be obtained by finding a better parameter setting, which is not trivial for RC. In our approach, a better setting is easier to find with fewer and more interpretable parameters. Here, we aim to visually highlight the possible extent of the differences between NVAR and rmESN, i.e., theoretically equivalent methods under different working points. To do so, we projected the data onto the first principal components using different kernel matrices (kPCA). When inspecting the projections for different datasets, we observed a few critical differences, predominantly for longer series. One example is illustrated in Fig. 5 (*CinCECGTorso* dataset). These appear in the form of a qualitatively

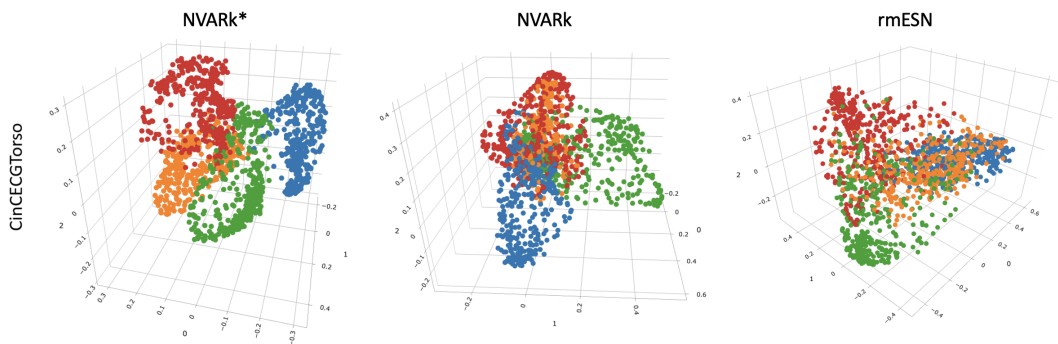

Figure 5: kPCA visualization of the *CinCECGTorso* dataset. Projection onto the first three principal components obtained with different kernels. Different colors correspond to different classes.

better grouping and class separation in the projected space. Tracing back the origin of this improvement, e.g., a

better working point or a better resilience to noise, would represent interesting future work. For an alternative visualization, we display here the two-dimensional projection of the *CinCECGTorso* dataset using different kernels.

## D  Individual results for SVM classification of UTS

We report here the individual accuracies obtained under the SVM classification framework of univariate time series of Sec. 4.1. Each accuracy is obtained as a mean of 10 repetitions with 10 different random seeds. For readability, the standard deviation over the runs is not reported. Datasets interpolated to T=400, before being processed by GAK, are marked with the symbol (*).

Table 9: SVM classification of 130 UTS datasets, individual performance. Results are averaged across 10 different seeds.

|  | NVARk* | NVARk | rmESN | TCK | GAK | SINK |
|---|---|---|---|---|---|---|
| ACSF1 | 0.835 | 0.826 | 0.792 | 0.646 | 0.396 | 0.800 |
| AbnormalHeartbeat | 0.707 | 0.737 | 0.688 | 0.733 | 0.716(∗) | 0.730 |
| Adiac | 0.749 | 0.709 | 0.638 | 0.623 | 0.598 | 0.731 |
| AllGestureWiimoteX | 0.673 | 0.676 | 0.608 | 0.471 | 0.527(∗) | 0.584 |
| AllGestureWiimoteY | 0.733 | 0.631 | 0.693 | 0.444 | 0.548(∗) | 0.651 |
| AllGestureWiimoteZ | 0.674 | 0.625 | 0.591 | 0.452 | 0.505(∗) | 0.622 |
| ArrowHead | 0.823 | 0.834 | 0.577 | 0.672 | 0.794 | 0.842 |
| BME | 0.973 | 0.927 | 0.983 | 0.967 | 0.953 | 0.971 |
| Beef | 0.847 | 0.590 | 0.783 | 0.610 | 0.897(∗) | 0.933 |
| BeetleFly | 0.800 | 0.950 | 0.880 | 0.645 | 0.850(∗) | 0.850 |
| BirdChicken | 0.850 | 1.000 | 0.785 | 0.590 | 0.700(∗) | 0.800 |
| CBF | 0.979 | 0.958 | 0.894 | 0.968 | 0.925 | 0.994 |
| Car | 0.833 | 0.733 | 0.612 | 0.693 | 0.825(∗) | 0.800 |
| CatsDogs | 0.737 | 0.726 | 0.709 | 0.663 | 0.626(∗) | 0.725 |
| Chinatown | 0.950 | 0.962 | 0.944 | 0.985 | 0.973 | 0.969 |
| ChlorineConcentration | 0.868 | 0.701 | 0.847 | 0.568 | 0.601 | 0.879 |
| CinCECGTorso | 0.914 | 0.849 | 0.834 | 0.850 | 0.888(∗) | 0.749 |
| Coffee | 0.964 | 0.964 | 1.000 | 0.971 | 0.968 | 1.000 |
| Computers | 0.776 | 0.685 | 0.712 | 0.658 | 0.527(∗) | 0.595 |
| CricketX | 0.697 | 0.645 | 0.649 | 0.618 | 0.681 | 0.744 |
| CricketY | 0.682 | 0.554 | 0.586 | 0.656 | 0.635 | 0.723 |
| CricketZ | 0.697 | 0.680 | 0.656 | 0.647 | 0.658 | 0.761 |
| Crop | 0.731 | 0.703 | 0.753 | 0.729 | *time limit* | *time limit* |
| DiatomSizeReduction | 0.968 | 0.912 | 0.858 | 0.911 | 0.964 | 0.977 |
| DistalPhalanxOutlineAgeGroup | 0.763 | 0.741 | 0.718 | 0.750 | 0.705 | 0.719 |
| DistalPhalanxOutlineCorrect | 0.779 | 0.775 | 0.733 | 0.787 | 0.740 | 0.764 |
| DistalPhalanxTW | 0.705 | 0.676 | 0.644 | 0.654 | 0.675 | 0.655 |
| DodgerLoopDay | 0.462 | 0.392 | 0.448 | 0.679 | 0.635 | 0.575 |
| DodgerLoopGame | 0.790 | 0.776 | 0.774 | 0.825 | 0.871 | 0.868 |
| DodgerLoopWeekend | 0.957 | 0.920 | 0.914 | 0.986 | 0.986 | 0.978 |
| ECG200 | 0.880 | 0.890 | 0.877 | 0.895 | 0.895 | 0.846 |
| ECG5000 | 0.940 | 0.940 | 0.940 | 0.938 | 0.939 | 0.943 |
| ECGFiveDays | 0.984 | 0.995 | 0.878 | 0.831 | 0.997 | 1.000 |
| EOGHorizontalSignal | 0.566 | 0.573 | 0.538 | 0.419 | 0.422(∗) | 0.512 |
| EOGVerticalSignal | 0.479 | 0.453 | 0.283 | 0.323 | 0.373(∗) | 0.417 |
| Earthquakes | 0.748 | 0.748 | 0.748 | 0.748 | 0.750(∗) | 0.748 |
| ElectricDevices | 0.654 | 0.650 | 0.652 | 0.643 | *time limit* | *time limit* |
| EthanolLevel | 0.579 | 0.360 | 0.587 | 0.382 | 0.607(∗) | 0.562 |
| FaceAll | 0.772 | 0.775 | 0.856 | 0.749 | 0.884 | 0.768 |
| FaceFour | 0.784 | 0.540 | 0.519 | 0.925 | 0.857 | 0.830 |
| FacesUCR | 0.903 | 0.811 | 0.808 | 0.852 | 0.890 | 0.895 |
| FiftyWords | 0.562 | 0.565 | 0.659 | 0.676 | 0.720 | 0.751 |

|  | NVARk* | NVARk | rmESN | TCK | GAK | SINK |
|---|---|---|---|---|---|---|
| Fish | 0.960 | 0.966 | 0.811 | 0.769 | 0.857(∗) | 0.906 |
| FordA | 0.982 | 0.928 | 0.945 | 0.674 | 0.910(∗) | 0.925 |
| FordB | 0.868 | 0.800 | 0.825 | 0.583 | 0.759(∗) | 0.777 |
| FreezerRegularTrain | 0.989 | 0.956 | 0.964 | 0.963 | 0.985 | 0.980 |
| FreezerSmallTrain | 0.836 | 0.905 | 0.822 | 0.743 | 0.691 | 0.700 |
| Fungi | 0.656 | 0.952 | 0.866 | 0.930 | 0.982 | 1.000 |
| GestureMidAirD1 | 0.662 | 0.591 | 0.535 | 0.448 | 0.535 | 0.554 |
| GestureMidAirD2 | 0.546 | 0.565 | 0.452 | 0.295 | 0.538 | 0.483 |
| GestureMidAirD3 | 0.238 | 0.248 | 0.204 | 0.177 | 0.290 | 0.215 |
| GesturePebbleZ1 | 0.773 | 0.573 | 0.591 | 0.739 | 0.693(∗) | 0.725 |
| GesturePebbleZ2 | 0.722 | 0.551 | 0.594 | 0.565 | 0.513(∗) | 0.618 |
| GunPoint | 0.987 | 1.000 | 0.963 | 0.961 | 0.960 | 0.980 |
| GunPointAgeSpan | 0.972 | 0.994 | 0.979 | 0.971 | 0.946 | 0.936 |
| GunPointMaleVersusFemale | 0.975 | 0.997 | 0.992 | 0.984 | 0.996 | 0.997 |
| GunPointOldVersusYoung | 1.000 | 1.000 | 1.000 | 1.000 | 1.000 | 1.000 |
| Ham | 0.717 | 0.739 | 0.707 | 0.730 | 0.748(∗) | 0.714 |
| HandOutlines | 0.948 | 0.945 | 0.866 | 0.884 | 0.917(∗) | 0.914 |
| Haptics | 0.525 | 0.399 | 0.437 | 0.444 | 0.445(∗) | 0.445 |
| Herring | 0.625 | 0.634 | 0.594 | 0.577 | 0.594(∗) | 0.605 |
| HouseTwenty | 0.916 | 0.923 | 0.900 | 0.893 | 0.785(∗) | 0.815 |
| InlineSkate | 0.543 | 0.470 | 0.443 | 0.310 | 0.340(∗) | 0.401 |
| InsectEPGRegularTrain | 1.000 | 1.000 | 1.000 | 1.000 | 1.000(∗) | 0.956 |
| InsectEPGSmallTrain | 1.000 | 1.000 | 1.000 | 1.000 | 1.000(∗) | 0.936 |
| InsectWingbeatSound | 0.312 | 0.372 | 0.515 | 0.638 | 0.632 | 0.617 |
| ItalyPowerDemand | 0.963 | 0.952 | 0.952 | 0.950 | 0.959 | 0.949 |
| LargeKitchenAppliances | 0.755 | 0.690 | 0.887 | 0.552 | 0.512(∗) | 0.794 |
| Lightning2 | 0.738 | 0.725 | 0.811 | 0.715 | 0.797(∗) | 0.820 |
| Lightning7 | 0.630 | 0.612 | 0.759 | 0.681 | 0.699 | 0.781 |
| Mallat | 0.959 | 0.936 | 0.909 | 0.939 | 0.924(∗) | 0.937 |
| Meat | 0.950 | 0.760 | 0.912 | 0.952 | 0.930(∗) | 0.933 |
| MedicalImages | 0.713 | 0.742 | 0.726 | 0.698 | 0.761 | 0.740 |
| MelbournePedestrian | 0.931 | 0.905 | 0.950 | 0.938 | 0.935 | 0.933 |
| MiddlePhalanxOutlineAgeGroup | 0.591 | 0.630 | 0.578 | 0.574 | 0.631 | 0.631 |
| MiddlePhalanxOutlineCorrect | 0.866 | 0.834 | 0.779 | 0.786 | 0.632 | 0.811 |
| MiddlePhalanxTW | 0.552 | 0.558 | 0.560 | 0.588 | 0.580 | 0.594 |
| MixedShapesRegularTrain | 0.946 | 0.936 | 0.913 | 0.874 | 0.903(∗) | 0.923 |
| MixedShapesSmallTrain | 0.913 | 0.867 | 0.798 | 0.817 | 0.846(∗) | 0.885 |
| MoteStrain | 0.812 | 0.792 | 0.694 | 0.882 | 0.861 | 0.870 |
| NonInvasiveFetalECGThorax1 | 0.930 | 0.924 | 0.935 | 0.890 | 0.922(∗) | 0.937 |
| NonInvasiveFetalECGThorax2 | 0.943 | 0.935 | 0.938 | 0.903 | 0.942(∗) | 0.950 |
| OSULeaf | 0.955 | 0.909 | 0.845 | 0.475 | 0.585(∗) | 0.707 |
| OliveOil | 0.867 | 0.867 | 0.793 | 0.853 | 0.407(∗) | 0.700 |
| PLAID | 0.737 | 0.749 | 0.521 | 0.424 | 0.396(∗) | 0.393 |
| PhalangesOutlinesCorrect | 0.826 | 0.834 | 0.784 | 0.797 | 0.664 | 0.798 |
| Phoneme | 0.297 | 0.267 | 0.335 | 0.221 | 0.117(∗) | 0.193 |
| PickupGestureWiimoteZ | 0.600 | 0.560 | 0.680 | 0.550 | 0.642 | 0.614 |
| PigAirwayPressure | 0.856 | 0.702 | 0.226 | 0.109 | 0.122(∗) | 0.216 |
| PigArtPressure | 0.954 | 0.962 | 0.514 | 0.210 | 0.248(∗) | 0.779 |
| PigCVP | 0.926 | 0.885 | 0.815 | 0.152 | 0.173(∗) | 0.601 |
| Plane | 1.000 | 1.000 | 0.981 | 0.990 | 0.973 | 0.968 |
| PowerCons | 0.883 | 0.873 | 0.943 | 1.000 | 1.000 | 0.983 |
| ProximalPhalanxOutlineAgeGroup | 0.820 | 0.849 | 0.849 | 0.853 | 0.840 | 0.837 |
| ProximalPhalanxOutlineCorrect | 0.900 | 0.900 | 0.858 | 0.855 | 0.778 | 0.849 |
| ProximalPhalanxTW | 0.771 | 0.795 | 0.763 | 0.801 | 0.802 | 0.797 |
| RefrigerationDevices | 0.541 | 0.523 | 0.542 | 0.512 | 0.419(∗) | 0.483 |
| Rock | 0.644 | 0.686 | 0.514 | 0.546 | 0.618(∗) | 0.508 |
| ScreenType | 0.552 | 0.416 | 0.483 | 0.402 | 0.455(∗) | 0.423 |

|  | NVARk* | NVARk | rmESN | TCK | GAK | SINK |
|---|---|---|---|---|---|---|
| SemgHandGenderCh2 | 0.871 | 0.840 | 0.829 | 0.886 | 0.912(∗) | 0.909 |
| SemgHandMovementCh2 | 0.717 | 0.562 | 0.600 | 0.756 | 0.768(∗) | 0.829 |
| SemgHandSubjectCh2 | 0.816 | 0.737 | 0.711 | 0.841 | 0.893(∗) | 0.913 |
| ShakeGestureWiimoteZ | 0.800 | 0.800 | 0.842 | 0.634 | 0.704 | 0.680 |
| ShapeletSim | 0.950 | 0.950 | 0.871 | 0.537 | 0.462(∗) | 0.800 |
| ShapesAll | 0.867 | 0.843 | 0.782 | 0.745 | 0.774(∗) | 0.834 |
| SmallKitchenAppliances | 0.656 | 0.607 | 0.706 | 0.630 | 0.508(∗) | 0.684 |
| SmoothSubspace | 0.960 | 0.945 | 0.903 | 0.980 | 0.943 | 0.900 |
| SonyAIBORobotSurface1 | 0.945 | 0.930 | 0.807 | 0.936 | 0.708 | 0.894 |
| SonyAIBORobotSurface2 | 0.918 | 0.928 | 0.834 | 0.829 | 0.816 | 0.879 |
| StarLightCurves | 0.976 | 0.966 | 0.969 | 0.937 | 0.946(∗) | 0.959 |
| Strawberry | 0.951 | 0.965 | 0.962 | 0.966 | 0.924 | 0.951 |
| SwedishLeaf | 0.922 | 0.929 | 0.916 | 0.907 | 0.895 | 0.930 |
| Symbols | 0.911 | 0.945 | 0.950 | 0.877 | 0.897 | 0.913 |
| SyntheticControl | 0.980 | 0.963 | 0.909 | 0.990 | 0.979 | 0.994 |
| ToeSegmentation1 | 0.904 | 0.970 | 0.890 | 0.615 | 0.635 | 0.864 |
| ToeSegmentation2 | 0.862 | 0.883 | 0.814 | 0.778 | 0.795 | 0.865 |
| Trace | 1.000 | 1.000 | 1.000 | 0.931 | 0.922 | 0.960 |
| TwoLeadECG | 0.978 | 1.000 | 0.871 | 0.692 | 0.861 | 0.978 |
| TwoPatterns | 0.988 | 0.822 | 0.767 | 0.997 | 0.953 | 0.999 |
| UMD | 0.993 | 0.986 | 0.856 | 0.963 | 0.961 | 0.932 |
| UWaveGestureLibraryAll | 0.967 | 0.706 | 0.727 | 0.946 | 0.958(∗) | 0.965 |
| UWaveGestureLibraryX | 0.697 | 0.698 | 0.631 | 0.766 | 0.772 | 0.792 |
| UWaveGestureLibraryY | 0.624 | 0.623 | 0.630 | 0.668 | 0.682 | 0.689 |
| UWaveGestureLibraryZ | 0.704 | 0.695 | 0.621 | 0.684 | 0.712 | 0.735 |
| Wafer | 0.993 | 0.992 | 0.995 | 0.993 | 0.995 | 0.998 |
| Wine | 0.722 | 0.815 | 0.794 | 0.707 | 0.606 | 0.667 |
| WordSynonyms | 0.434 | 0.440 | 0.572 | 0.596 | 0.647 | 0.669 |
| Worms | 0.779 | 0.822 | 0.808 | 0.569 | 0.621(∗) | 0.670 |
| WormsTwoClass | 0.805 | 0.827 | 0.805 | 0.568 | 0.662(∗) | 0.717 |
| Yoga | 0.831 | 0.822 | 0.793 | 0.795 | 0.845(∗) | 0.877 |

