# OpenReview forum: "Time Series Kernels based on Nonlinear Vector AutoRegressive Delay Embeddings"
_NeurIPS.cc/2023/Conference — NeurIPS 2023 poster_

### Official Review · Reviewer_XEWY · 2023-07-03

**Soundness:** 3 good
**Presentation:** 3 good
**Contribution:** 3 good
**Rating:** 7
**Confidence:** 3

**Summary:**

This focus of this paper on a new type of kernel for reservoir computing (using nonlinear vector autoregressive delay embeddings) that is faster and more accurate than related kernels.

The three main contributions
* Introduction of a new kernel for time-series modeling
* State of the art results on uni- and multimodal problems on real-world datasets (related work has used synthetic datasets.)
* Theoretical connections to Taken's theorem and the field of state space reconstruction.

There are only a small number of (intuitive) hyper parameters for tuning the model. While this approach does not seem suitable for large datasets (where deep learning-based solutions reign), it appears to be a nice and efficient solution for smaller time-series problems.

**Strengths:**

Overall the paper is well written. I don't have a deep understanding of reservoir computing but I think they authors do a nice job of connecting different technical areas. If I was working on problems in this area I would be strongly interested in building a deeper understanding of this paper.

The authors do a nice job of contextualizing the work, especially given that this is deviates from many recent deep learning oriented time-series papers.

The core idea behind the kernel is elegant and clearly has a large impact on efficiency of the approach.

I appreciate the discussions on asymptotic complexity and the qualitative notes and visualizations.

**Weaknesses:**

It looks like the authors only use 122 UCR unimodal datasets but my understanding is that the most recent UCR benchmark (since 2018) has been using 128 unimodal datasets. Is there a good reason for this discrepancy? Similarly, the UEA corpus should have 30 datasets whereas only 22 are used in this paper.

Minor: it's challenging to interpret the graphs (especially Fig 2) when printed in black and white.

**Questions:**

At the end of section 2, there is discussion on the lack of stability of techniques for reservoir computing. Perhaps I'm being naive, but this seems obvious given that many of the parameters (e.g., W_in and A) are randomly initialized and not optimized over -- this means that parameters that are updated (e.g., W_out and c) need to compensate for potentially poor choices of W_in and A. I know this isn't something you introduced, but generally speaking why is this a reasonable idea?

The approach is described as "a simple concatenation of the input series with time-delayed copies and nonlinear functionals." Could this approach simply be implemented using convolutions (with dilations)? Would there be a more straightforward way of writing out this math in this case?

**Limitations:**

This paper is more theoretical and does not touch on noteworthy societal impacts.

---

> ### Author Rebuttal · Authors · 2023-08-08
>
> We warmly thank reviewer $\color{orange}{\textbf{XEWY}}$ for the thorough review and positive feedback.
> We really appreciate the favorable comments on clarity of the writing and elegancy of the underlying idea, as well as positive remarks on how we bring together different research areas.
> As for the raised concerns and questions, individual points are addressed below.
>
>
> **W1. Is there a reason for benchmarking on a subset of datasets?**\
> Among the univariate datasets, we have excluded 8 datasets with a high $N_{classes} / N_{train}$, which are then incompatible with our choice of stratified cross-validation for the SVM.
> Results for those datasets can still be obtained by considering a shuffle split rather than stratified, which we report in Tab. 2 (attached pdf).
> Among multivariate ones, we have excluded 6 high-dimensional datasets for which the application of the NVAR kernel is not appropriate, and 5 datasets for which SVM accuracy was $\ll 50 \%$ for all approaches, which indicates that kernel methods are, in general, not a suitable solution.
> Thanks to the reviewer's comment, we now include 3 additional multivariate datasets that we did not originally consider.
> Corresponding results are shown in Tab. 2.
> Average accuracy for the additional univariate and multivariate datasets indicates superior performance of NVARk with respect to the baselines.
> All results have been integrated into the experimental section of the manuscript.
>
>
> **W2. Figures in black and white**\
> We thank the reviewer for pointing this out, we have included a line where, for interpretation of colors, we refer the reader to the web version of the article.
>
>
> **Q1. Why are unoptimized parameters in RC a reasonable idea?**\
> One of the greatest advantages of keeping the parameters of the recurrent update unoptimized is to avoid exploding or shrinking the gradient when backpropagating through time.
> This is a complementary solution to maintaining the memory unaltered with the popular gating mechanisms, e.g. LSTMs and GRUs, but much more efficient.
> Furthermore, despite their randomness, the reservoir dynamics have the potential to amplify the relevant features in the input data, actually reducing the burden on the linear readout.
> However, if the choice of the hyperparameters (controlling such dynamics) is poor, an adverse effect is obtained.
>
>
> **Q2. Could the approach be rewritten using dilated convolutions?**\
> It is interesting to reason about this.
> Convolutions can be declined into products between the input and a sliding filter or between two delayed signals.
> Regarding the first type, in our approach, there is no weighting of the input dimensions until the final readout.
> Despite this layer certainly involving some terms of the form $w_k x_{t-j}$, non-linear terms are different.
> For what concerns these non-linear terms, they are products between the input and its lags.
> It is then definitely possible to think of a rewriting in terms of causal convolutions of the input with itself, despite the advantages not being fully clear to us.
> For a full rewriting, one can think of merging both considerations.
> However, at first sight, this seems far from straightforward.
>
> We hope to have clarified all of the reviewer’s concerns and are happy to provide further details.

---

> > ### Comment · Reviewer_XEWY · 2023-08-16
> >
> > Thanks for the depth of thinking in all of your responses. Based on the responses here and the other reviews/comments I am increasing my score to 'accept.'

---

### Official Review · Reviewer_Nbgs · 2023-07-06

**Soundness:** 3 good
**Presentation:** 3 good
**Contribution:** 2 fair
**Rating:** 4
**Confidence:** 4

**Summary:**

This paper presents a kernel that can be applied to time series - both
univariate and multivariate - that draws inspiration from reservoir
computing.  Practically, it constructs an expanded time series by
sampling lags and then computing polynomial combinations of the
original and lagged data through time.  Experiments show that the
kernel, when combined with an SVM, is more accurate and more efficient
than a number of relevant competitors.


**Strengths:**

This paper pulls on an interesting thread of work in reservoir
computing (RC) that makes connections between RC and nonlinear vector
autoregressive models (NVARs).  RC can be computationally expensive,
and the NVAR approach simply combines time series with lagged copies
and nonlinear functionals, like products.  The paper further connects
the use of lags to Takens' theorem about reproducing noise-free
nonlinear dynamics through state space reconstruction using lagged
observations.  The discussion about the various connections to this
work is interesting.

The paper is clear and relatively easy to read and understand.  RC and
NVAR are summarized in a useful way without getting bogged down in
details, but with enough information to follow the presentation.  The
figures are also very helpful in understanding the paper.

One potential weakness of the approach is the relatively large number
of hyperparameters, including the number and size of the lags, the
order of the polynomial regression, and the embedding dimension.  The
paper, though, does a good job presenting and arguing for heuristic
choices for these parameters that seem to perform well in practice.

Finally, empirical results suggest that the proposed approach is as
accurate as the state of the art and more efficient.


**Weaknesses:**

The primary contribution of the paper is a kernel that is accurate and
fast to compute.  There are claims in the introduction where the
authors suggest that the kernel is more interpretable than competitors
though there is no discussion of that in the remainder of the paper.

The approach is conceptually simple, which is good - fit a model to an
augmented time series (lags and products) and use the model parameters
as a vector-based representation of the time series.  The
hyperparameter choices - of which there are many - are made
heuristically.  Some indication of the sensitivity of the approach to
variation in those choices would be good.  That is, are the heuristics
a result of just those values working empirically; or is the method
highly sensitive to those choices?

The complexity analysis would benefit from doing a direct comparison.
That section claims that the proposed methods does one expensive
computation (fitting a model) per, compared to other methods that do
an expensive computation per pair of instances.  What is the total
complexity of the approaches?  If competing approaches have less
expensive per-pair cost they may still be more efficient overall.

It's unclear what the kPCA plots add to the paper.  What dataset is
represented in the figure?  Why did you choose that dataset?  I'm sure
there are other datasets for which the separation for rmESN is better
than for NVARk.

An ablation study in which the MTS were augmented with just lags or
with just products would help tease out which element is most
important.  Takens says nothing about nonlinear features.



**Questions:**

(1) How sensitive is the approach to hyperparameter choice, both in
terms of complexity and accuracy?

(2) Are both lags and products important for accuracy?  Why or why not?


**Limitations:**

There is no such discussion in the paper.

---

> ### Author Rebuttal · Authors · 2023-08-08
>
> We warmly thank reviewer $\color{magenta}{\textbf{Nbgs}}$ for the extensive review and constructive feedback.
> Among all, we really appreciate the kind words on the clarity of our paper and figures, as well as showing interest in how we connect our work to different research areas.
> As for the raised concerns and questions, individual points are addressed below.
>
>
> **W1. There is no discussion of interpretability in the remainder of the paper**\
> For a discussion on our interpretability claims, we kindly refer the reviewer to our common response \#1.
>
>
> **W2. How sensitive is the approach to hyperparameter choice?**\
> In addressing this, we split hyperparameters into classes, as we believe a separate discussion is needed for $k$ and $s$.
>
> -
>    - $\bar{d}_r$: as we show in Fig. 1 (top) (attached pdf) average performance for multivariate datasets are not considerably impacted in the range [75, 150]. As expected, we observe a significant drop of performance for high values (redundancy).
>    - $\lambda_{ridge}$: we show in Fig. 1 (bottom-left) little sensitivity in the region $[5, 50]$. Results are averaged across a pool of 20 randomly sampled univariate datasets. The employed OCReP method outperforms the best choice by $\sim 2\%$, as it adapts the value to each individual dataset.
>    - $\gamma_{rbf}$: in Fig. 1 (bottom-right), we range different multiplicative factors and show little sensitivity in a neighborhood of the unity.
>
> - The sensitivity to $k$ and $s$ strongly depends on the dataset, although it generally tends to be high.
> In response, we introduce our dataset-specific heuristic adapting ideas from a precedent study [a], which we find works well in practice.
> A reasonable idea of the average sensitivity to these parameters can be obtained by comparing the performance of NVAR, where k and s are chosen by heuristic, to the performance of NVARk*, where k and s are optimized.
>     Unfortunately, understanding sensitivity by considering other settings is challenging, as the existing literature predominantly focuses on the noise-free case.
>     Furthermore, there is no strong evidence for the wider applicability of such approaches outside dynamical systems (see **Q1** from reviewer $\color{red}{\textbf{RJLc}}$).
>
>
>
> **W3. What is the total complexity of the approaches?**\
> As evidenced in Section 3.4, NVARk exhibits an overall complexity of $\mathcal{O}(NT\bar{d}^2_r)$.
> SINK and GAK report $\mathcal{O}(T \log(T))$ and $\mathcal{O}(\min(T_1, T_2))$ respectively, which do not counterbalance their $\mathcal{O}(N^2)$ nature.
> A separate discussion is needed for TCK, which gets around its cubic complexity by putting an upper threshold on the length of the sampled time segments ($\mathcal{O}(T_{max}^3)$).
> As for dimensionality $\bar{d}_r$, we treat it as a constant and do not discuss complexity in connection to this parameter.
>
>
> **W4. It’s unclear what the kPCA plots add to the paper**\
> Despite the underlying equivalence between NVAR and simple RC, better performance can be obtained by finding a better parameter setting, which is not trivial for RC.
> In our approach, a better setting is easier to find with fewer and more interpretable parameters.
> We mean to use the kPCA example (*CinCECGTorso* dataset) to visually highlight the possible extent of the differences between NVAR and rmESN, i.e. theoretically equivalent methods under different working points.
> As a side note, it highlights that a kernel can be used for visualization.
>
>
>
> **W5. Are both lags and products important for accuracy?**\
> This is an interesting ablation study that has not been investigated in our work and previous works to the best of our knowledge.
> We provide results for such additional experiments in Tab. 1 (attached pdf).
> We separately consider the case of fitting directly on the input (no concatenation), sampling linear terms only, and sampling non-linear terms only.
> As a general trend, we observe that using all terms leads to the best results, followed by \textit{non-linear} and \textit{linear}.
> Note also that all configurations perform better than fitting directly on the input series.
> In the univariate case, the \textit{linear} variant tends to underperform, as the total number of concatenated dimensions is very small (equal to $k$).
> For multivariate datasets, the two variants perform very similarly, as all lags of attributes may already include a considerable number of dimensions.
> Both results are also in line with our interpretation using the generalized Taken's theorem.
> Inspired by this study, we have also tested an additional variant in which we add all linear terms, and then fill remaining dimensions (up to $\bar{d}_r$) by sampling nonlinear ones.
> Interestingly, this leads to even better results and a reduction in variance.
> We believe this compensates for the imbalance between all possible linear and non-linear terms, while also reducing potential noise from uncorrelated dimensions.
> We sincerely thank the reviewer for raising this question, which led to additional interesting results and the integration of this comprehensive analysis into the main experimental section.
>
> **W6. There is no discussion of limitations **\
> We kindly refer the reviewer to lines 330-331, where the main limitation is mentioned in the manuscript.
> Please also find a further discussion in our common response \#2.
>
> We hope to have clarified all of the reviewer’s concerns and are happy to provide further details.
>
> **References:**\
> [a] Paparrizos et al., State space reconstruction parameters in the analysis of chaotic time series - the role of the time window length, Physica D, 1996.

---

### Official Review · Reviewer_8sG2 · 2023-07-19

**Soundness:** 4 excellent
**Presentation:** 4 excellent
**Contribution:** 3 good
**Rating:** 7
**Confidence:** 4

**Summary:**

The authors introduce a new time series kernel based on Nonlinear Vector AutoRegressive (NVAR) processes, following recent literature on its equivalence to reservoir dynamics. The kernel operates on time delay embeddings and enables the computation of similarity between time series with different lengths. The proposal is paired with SVM models and evaluated in classification tasks for both univariate and multivariate cases. The obtained accuracy and speed results are promising when compared to other kernel methods and a reservoir computing alternative.

**Strengths:**

The manuscript is well written and organized. The motivation is clear and the tackled problem (building kernels for time series data) is relevant to the community. The discussion of the related methods is comprehensive, although I have missed a nod to Gaussian process models (see 'Questions' section).

The main proposal seems to merge the contributions by Bianchi et al. (2020) and Bollt (2021), evaluating the NVAR equivalence to RC in the task of time series classification. The simple to understand heuristics that form the NVARk is a relevant strength of the work.

The main proposal, NVARk, is explained in detail and the provided source code that implements Algorithm 1 is easy to follow. I also praise the provided supplementary material and the qualitative result for KPCA visualization.

**Weaknesses:**

The performed experiments are quite extensive, but the results presentation can be improved. For instance, Table 1 could also present the average rank of each method and be paired with boxplots of accuracy results across the 122 univariate datasets. Beside the per dataset results, Table 2 should also include the metrics considered in Table 1 (rank, average score), for the sake of homogeneity.

**Questions:**

- I believe a more thorough discussion on how Taken's theorem holds with random lagged dimension dropping (Eq. (5)) is required. Perhaps some of the content of the supplementary material could be brought to the main text.

- In practice, is there any interplay between the choice of $k$ lagged steps and polynomial order $n$? In which scenarios n > 2 would be useful?

- How the proposed method behave for larger dimensional datasets? It would be interesting to include the dataset dimension in Table 2.

- Lines 315-316: "NVARk can, in fact, get around both issues with its non-recursive structure and the absence of a training phase." -> I suppose the ridge regression step and the paired SVM are not considered here, which should be made clearer.

- Since the tuned NVARk* usually presents even better results, it would be interesting to verify its trade-off by including it in the experiments of Section 4.3 (Execution time and scalability).

- It would be valuable to comment about the use of the introduced NVARk in the context of Gaussian processes models. Maybe a brief discussion on how it would compare with other GP approaches for sequences [1,2,3,4] could be included.

References

[1] Frigola-Alcade R. and Rasmussen, C. Integrated pre-processing for Bayesian nonlinear system identification with Gaussian processes. IEEE CDC, 2013.

[2] Mattos, C. et al. Recurrent Gaussian processes. ICLR, 2016.

[3] Al-Shedivat, M. et al. Learning scalable deep kernels with recurrent structure. JMLR, 2017.

[4] Toth, C. and Oberhauser H. Bayesian learning from sequential data using Gaussian processes with signature covariances. ICML, 2020.

**Limitations:**

The manuscript states sufficiently its limitations.

---

> ### Author Rebuttal · Authors · 2023-08-08
>
> We warmly thank reviewer $\color{green}{\textbf{8sG2}}$ for the extensive review and positive feedback.
> We particularly appreciate the positive remarks on clarity of the presentation and soundness of the work, which we are happy to see reflected in the scores.
> In addition, we value your comment on the relevancy of kernels for time series analysis.
> As for the raised concerns and questions, individual points are addressed below.
>
>
> **W1. Results presentation can be improved**\
> We agree, our initial presentation choices were mostly dictated by the space constraint.
> In the new version of the manuscript, we now provide the average rank metric, which reflects what is observed in AVERAGE SCORE:
>
> AVERAGE RANK: NVARk = 2.1, SINK = 2.2, rmESN = 2.6, TCK = 2.8
>
> We now also provide consistent metrics for univariate and multivariate data, i.e.
>
> AVERAGE SCORE (multivariate): NVARK* = 0.890; NVARk = 0.877; rmESN = 0.857; TCK = 0.842\
> AVERAGE RANK (multivariate): NVARk = 1.62; rmESN = 2.12; TCK = 2.25\
> N FIRST RANKED (multivariate): NVARk = 10; rmESN = 2; TCK = 4
>
> As for the boxplot, we argue that it might confuse the reader as we do not claim a statistically significant difference with respect to SOTA.
>
>
> **Q1. More thorough discussion on how Taken’s theorem holds with random lagged dimension dropping is required**\
> We agree that the paper would benefit from a more thorough introduction to state space reconstruction for the unfamiliar reader.
> Takens' theorem establishes an upper limit on the necessary concatenated lag count that is necessary to form a valid equivalent of the generating dynamical system.
> In Takens' formulation, there is no importance bias towards any specific lag and their number is the only relevant factor.
> In essence, this rationale underpins our adoption of random lag dropping, as long as 'enough' concatenated dimensions remain.
> In our next revised version, we will prioritize integrating the main text with material from Appendix A.
>
>
> **Q2. Is there any interplay between the choice of lagged steps and polynomial order?**\
> There is, but we could not find any use for it.
> The polynomial order ($n$) regulates the degree of interactions between dimensions.
> With $n=2$, NVAR captures mutual correlations between attributes and auto-correlations across lags.
> With $n=3$, three-way interactions are considered as well, such as dimensions being correlated only if a third one exhibit certain patterns.
> If these are important for the dataset at hand, then $n=3$ could, in principle, be considered.
> Meanwhile, $k$ influences the model's memory span.
> In words, more lags would enable the model to explore these interactions further back in time.
> However, in practice, relevant three-way interactions tend to be rare and our random sampling often leads to inadvertent spurious correlations (something that would appear even more often with a high $k$).
> As a result, we have never found any advantage in using $n > 2$.
>
>
> **Q3. How the proposed method behave for larger dimensional datasets?**\
> We kindly refer the reviewer to our common response \#2.
>
>
> **Q4. Clarify "absence of training phase"**\
> The reviewer is right in saying that ridge regression and SVM are not considered in the mentioned statement.
> At ll.315-316, we have substituted this terminology with a reminder of NVARk's linear complexity in the number of time series ($N$).
>
>
> **Q5. It would be interesting to verify trade-off of optimized NVARk**\
> The reported superior performance for NVARk* can be obtained with a median slowdown factor of $\sim25$ (parallelized grid search + 1 iteration with best parameters) with respect to the heuristic-based NVARk.
> Most unfavorable cases are high-T datasets.
> As for asymptotic behavior, this is trivially similar to NVARk scaled by the grid size.
> NVARk* also places favorably with respect to the considered baselines.
> We report here the median computation times:
> NVARk = $0.5 s$, NVARk* = $13.7 s$, SINK = $24.2 s$, rmESN = $38.3 s$, TCK = $44.9 s$.
> Interestingly, we also observed that NVARk* execution time is faster than 1 iteration of rmESN for 107/122 univariate datasets.
>
>
> **Q6. It would be valuable to comment about the use in the context of GP**\
> This is an interesting direction that can be further explored.
> We find these two directions appealing.
> As a first example, such MTS kernels can, in general, be used in multi-task GP, where they can be paired with a scalar kernel over time (like RBF) for simultaneous modeling of multiple time series:
> $$ k(t_1, t_2, i, j) =  RBF(t_1, t_2) \cdot K_{MTS}(i,j) $$
> e.g. spatio-temporal data or medical data for different patients.
> Secondly, and more specifically to NVARk, we recognize a compelling similarity between building our feature map (linear terms and polynomials) and different orders of information in signature transforms.
> As in a reference suggested by the reviewer, signatures are being employed as a building block in deep kernels to build larger GP models, which might pave the path for interesting future directions.
>
> We hope to have clarified all of the reviewer’s concerns and are happy to provide further details.

---

> > ### Comment · Reviewer_8sG2 · 2023-08-10
> >
> > I thank the authors for the answers and the additional results in the provided pdf. I suggest including the new figures and tables in the work's appendix. I believe the comments presented in the common response #2 is a valuable inclusion to the main text.
> >
> > Due to the clear responses and the additional discussion provided by the authors' rebuttal to all the reviews, I will increase my original score.

---

### Official Review · Reviewer_hHQP · 2023-07-19

**Soundness:** 3 good
**Presentation:** 3 good
**Contribution:** 3 good
**Rating:** 7
**Confidence:** 3

**Summary:**

This work proposes a feature extraction based on a non-linear vector autoregressive model (called NVAR kernel in the paper). The NVAR method constructs a deterministic feature matrix with the original input time series, lagged versions of the time series (parametrized by the lag and spacing parameters) and the products between lagged and unlagged dimensions, up to some polynomial order. The resulting feature map is then subsampled to obtain a random feature map. A linear readout layer performs next-step prediction on the embedding states and the resulting parameters form the feature map for a radial basis function kernel. The authors form a connection to state space reconstruction theory to justify their approach of subsampling by connecting it to state space reconstruction theory. The method is compared to other kernel choices on univariate (109) and multivariate (16) datasets for classification. The authors also demonstrate the scalability of their approach by measuring execution time and provide heuristics for hyperparameter choices.

**Strengths:**

Originality
The underlying ideas in this paper have been proposed before in the time series space (lags and polynomial features; randomized feature maps), but the combination proposed here is novel for time series classification. The NVAR formalism in time series classification is novel as well (to the best of my knowledge).

Quality
The algorithm choices presented in this paper are motivated by referring to previous work and connecting to state space reconstruction theory. While I’m not an expert on this, I appreciate the effort the authors made to justify the randomized subsampling step. The paper evaluates the proposed approach against other kernel choice baselines from different classes (reservoir, model-based, Fourier-based) univariate and multivariate datasets. The authors also analyze the execution time and scalability. I appreciate the efforts from the authors to provide heuristics for hyperparameter choices of their approach and showing the gain of additional cross-validation. I think the paper misses the comparison to a critical baseline, which I will discuss in the Weaknesses section.

Clarity
Overall, the paper was well written but I think the clarity could be improved by introducing some core concepts into the introduction to readers that are not familiar with reservoir computing. For example, the authors state in the introduction that reservoir computing has a “large set of hyperparameters” but do to mention what they mean. I think a bit more detail to introduce the reader would be helpful (although I would consider this largely a stylistic choice as most of this detail comes later in the paper).

Significance
Time series classification is an important application of time series research. Especially in the healthcare domain, time series classification methods have to run on devices with limited compute (smartphones or other edge devices). As such, a classification method that does only require minimal compute is significant. However, since this method is specific fairly specific to time series classification, the broader impact is limited.


**Weaknesses:**

I agree the rationale of the authors to select one representative baseline methods per category (reservoir, model based) etc. However, I think the the MINIROCKET method (Dempster et al. KDD 21: https://dl.acm.org/doi/10.1145/3447548.3467231) shares many properties of the proposed method (per time series computation is the most expensive step, suitable for downstream linear classification, low computational cost). Since this method provides high classification accuracy at low computational cost, I think it should be one of the baselines in the paper (it can also deal with uneven sequence length due to its final pooling operation). I would appreciate if the authors consider to compare against the method. At the very least, I would expect that this method is discussed in the related work (even though it is not from the reservoir computing literature).

**Questions:**

See Weaknesses.

**Limitations:**

Limitations are not discussed in the paper (or I missed it). A short discussion of limitations would be appreciated.

---

> ### Author Rebuttal · Authors · 2023-08-08
>
> We deeply thank reviewer $\color{blue}{\textbf{hHQP}}$ for the extensive review and positive feedback.
> We really appreciate the positive remarks on the general quality and clarity of our paper.
> Similarly, we value the highlighting of a form of novelty and significance.
> As for the raised concerns and questions, individual points are addressed below.
>
>
> **W1. A bit more detail to introduce the reader to RC would be helpful**\
> We agree that the paper would benefit from a more extensive introduction to RC and its hyperparameters, as their non-interpretability and large number partially motivate our approach.
> Our decision not to delve into extensive discussions originated from the need to adhere to the space limitations.
> As a middle ground, we now mention a few representative examples around l.120, i.e. spectral radius and input scaling (control non-linearity and relative effect of the current input as opposed to the history), and leaking rate (control speed of reservoir updates).
> We also refer the reader to Appendix B for a comprehensive list.
>
>
> **W2. Broader impact is limited**\
> While demonstrating versatility across tasks is valuable, we respectfully disagree that assessing one task suggests limited impact.
> Within the established literature, typical comparison of kernels uses classification only [a]-[c], but their application notably extends beyond that.
> As in Mikalsen et al. [d], we also present a snapshot of kPCA.
> Most importantly, we would like to emphasize that the numerical accuracy of TS classification constitutes only a part of our work.
> Rather, we anticipate an impact on RC representation learning, for which we demonstrate superior performance with higher efficiency and interpretability.
> Additionally, we demonstrate success of extending applicability of NVAR to noisy multivariate time series analysis, which we argue is of interest to both communities.
>
> **W3. MINIROCKET should be one of the baselines**\
> We thank the reviewer for bringing MINIROCKET to our attention, we have included it in the background section.
> Our choice of representative for SOTA was based on the most extensive evaluation of univariate TS metrics to date [e], from where SINK comes out as the strongest trade-off between accuracy and efficiency.
> We agree that MINIROCKET, as a representation learning method, can be used in conjunction with an RBF kernel, which would allow comparison.
> In fact, this line of work certainly warrants further investigation.
> However, we are unable to provide proper comparison due to the time constraints.
> On one hand, one should ensure fairness of the hyperparameter setting, with no bias towards the UCR classification task (note that all heuristics for NVARk, except arguably $\bar{d}_r$, are task agnostic).
> On the other, the NVARk processing is of a causal type, while our understanding is that MINIROCKET attends to both the past and future of each timestamp.
> As such, we believe that a bidirectional variant of NVAR representation would be better for such comparison.
> On a side note, the focus of our work is more on the expressive power of RC methods rather than TS classification.
>
>
> **W4. Limitations are not discussed**\
> We kindly refer the reviewer to ll.330-331, where the main limitation is mentioned in the manuscript.
> Please also find a further discussion in our common response \#2.
>
> We hope to have clarified all of the reviewer’s concerns and are happy to provide further details.
>
> **References:**\
> [a] Paparrizos et al., Grail: efficient time-series representation learning, VLDB Endowment, 2019.\
> [b] Cuturi and Doucet, Autoregressive Kernels for Time Series, arXiv, 2011.\
> [c] Baydogan et al., Time series representation and similarity based on local autopatterns, Data Mining and Knowledge Discovery, 2016.\
> [d] Mikalsen et al., Time series cluster kernel for learning similarities between multivariate time series with missing data, Pattern Recognition, 2017.\
> [e] Paparrizos et al., Debunking four long-standing misconceptions of timeseries distance measures, ACM SIGMOD, 2020.

---

> > ### Comment · Reviewer_hHQP · 2023-08-17
> >
> > I would like to thank the authors for their response in the rebuttal.
> >
> > > Broader Impact is Limited
> >
> > Thank you for your response. I agree with the authors that limiting to classification does not mean that the method cannot be applied to other tasks, it also does not demonstrate the effectiveness of the method for other tasks. Some recent papers on time series representation learning do demonstrate the effectiveness of their method on several tasks and therefore do suggest wider applicability (forecasting, anomaly detection, classification, see [1]-[2]). I agree that the method is not limited to classification, but the applicability to other tasks is also not demonstrated in this paper.
> >
> > Zhihan Yue, Yujing Wang, Juanyong Duan, Tianmeng Yang, Congrui Huang, Yu Tong, & Bixiong Xu (2021). TS2Vec: Towards Universal Representation of Time Series. In AAAI Conference on Artificial Intelligence.
> >
> > Ling Yang, & linda Qiao (2022). Unsupervised Time-Series Representation Learning with Iterative Bilinear Temporal-Spectral Fusion. In International Conference on Machine Learning.
> >
> > Thank you for addressing the my other points. Given the thorough response to other reviewers, I'm increasing my score. I would kindly ask the authors to reconsider the last sentence of the abstract that is arguably correct, but suggests to the reader that the proposed method is demonstrated on forecasting (which it is not in this paper and refers to related work in this space).

---

> > > ### Author Response · Authors · 2023-08-21
> > >
> > > We thank the reviewer for their support.
> > >
> > > We also welcome their suggestion and have modified the last sentence of the abstract with:
> > > "This further advances the understanding of RC representation learning models and extends the typical use of the NVAR framework to kernel design and representation of real-world time series data."

---

### Official Review · Reviewer_RJLc · 2023-07-24

**Soundness:** 3 good
**Presentation:** 3 good
**Contribution:** 3 good
**Rating:** 7
**Confidence:** 3

**Summary:**

In their paper, the authors propose a new method for deriving a kernel from time series data. They combine ideas from reservoir computing with nonlinear vector autoregressive models, based on recent theoretical work exploring their similarities.
The primary idea of the paper is to construct a kernel by modeling a time series' dynamics using a nonlinear vector autoregressive model first. The vectorized fit parameters are then used to constitute the kernel representation of the respective time series.
The authors investigate the efficacy of this kernel in SVM classification across a range of unimodal and multimodal time series. They demonstrate state-of-the-art performance when compared to three other approaches.

**Strengths:**

Overall, the paper is well-written and easy to follow. The main ideas and contributions are clearly delineated. Further, code is provided with the submission. The method manages to significantly reduce the number of hyperparameters needed to model dynamics compared, for instance, to RC approaches. It also achieves state-of-the-art performance on a range of unimodal and multimodal time series classification tasks, compared to other kernel based methods. It presents an interesting contribution that as far as I can judge is novel, and leverages a recent development in a related field to come up with a new application.

**Weaknesses:**

Comparison methods: If I understood correctly, no hyperparameters were tuned for the comparison methods in the experiment, while you optimized the delay-embedding parameters for your method. While I appreciate that a thorough hyperparameter tuning for some of the comparison methods can be computationally prohibitive and the smaller amount of hyperparameters is one of the strengths of your approach, it would make the comparisons fairer if some tuning took place, or if you explained in more detail why you didn't perform any tuning.

Line 217: "We propose the region 75 ≲ ¯dr ≲ 100 as a good, though non-optimal, setting" The cutoff around 75-100 is justified experimentally in the appendix. The phrase "non-optimal setting" is repeated in the appendix. What do you mean by "non-optimal" in this context? Why would you choose a non-optimal setting? While I understand that this cutoff is motivated by decreased performance due to redundancy, it seems highly dependent on the dataset, and I'm not sure I understand why this reduces performance for the MTS so drastically. Since this is one of the central components/hyperparameters of your algorithm, some more clarification would be useful here.

**Questions:**

Relationship to Taken's Theorem: The choice of methodology is explained with reference to Taken's delay embedding. However, the connection to Taken's theorem in some of the practical cases isn't entirely clear to me. In the original next-generation RC approach, chaotic attractors are considered, where Taken's Theorem applies, while in your setting, many of the time series used in the classification tasks are not derived from something that can be as straightforwardly modeled via an underlying attractor (e.g., Japanese Vowels). It might be interesting to investigate further for which types of time series your approach is particularly suited and outperforms other methods. It's also worth exploring whether this relates to the suitability of a dynamical systems description for the underlying dynamics.

l.56: The resulting model is more accurate, interpretable, and non-recurrent.
What does interpretable refer to in this context? The single k-PCA example does not make this point strong enough in my mind, and it is not apparent to me how the derived parameters are more interpretable than e.g. the parameters of an RC model.

Small typos:
226, implies
236: only the ridge
328: Computationally, it is exceptionally

**Limitations:**

The authors explicitly mention that high-dimensional datasets are the main weakness of their approach. I don't foresee any direct negative societal impact.

---

> ### Author Rebuttal · Authors · 2023-08-08
>
> We warmly thank reviewer $\color{red}{\textbf{RJLc}}$ for the thorough review and positive feedback.
> We really appreciate the kind words on the clarity of our paper, as well as the novelty and innovative utilization of recent developments from a related field.
> Lastly, we are also grateful for spotting a few typos.
> As for the raised concerns and questions, individual points are addressed below.
>
>
> **W1. No hyperparameters were tuned for the comparison methods**\
> This is only partially true.
> We indeed found fine-tuning baseline hyperparameters to be mostly computationally unfeasible and, considering NVARk's heuristic foundation, might also not yield a fair comparison.
> However, to the best of our knowledge, there are no rule-of-thumb guidelines facilitating the choice.
> Our choice of baseline hyperparameters is then based on what is provided in the authors' original papers, with the belief that suit their approaches well and are transferable, especially given that our task and benchmarking archives are similar (or a subset).
> When applicable, we consistently apply the same choices for both NVARk and rmESN, i.e. OCReP optimization for $\lambda_{ridge}$ and heuristic for $\gamma_{rbf}$.
> Also, note that our choice for the hidden dimensionality ($\bar{d}_r = 75$) is optimized for rmESN in the authors' paper (see Supplementary Material for [a]), and is not optimal for NVARk.
> Finally, we would like to emphasize that performance for optimized NVARk (NVARk*) is reported separately.
>
>
> **W2. Why would you choose a non-optimal setting for $\bar{d}_r$?**\
> In line with the scope of our work, the main motivation for using $\bar{d}_r = 75$ (l.252) is to prioritize a fair comparison with rmESN.
> Our term "non-optimal" refers to the lack of fine-tuning of $\bar{d}_r$.
> The fact that this value lies within the range $70-100$, where we empirically observe an absence of feature redundancy, made us accept this value.
> We have made this clearer at ll.213-217, with a recall at l.252.
> As for the observed performance drop after this range, please find a more extensive discussion in our common response \#2.
>
>
> **Q1. It’s worth exploring whether performance relates to the suitability of a dynamical systems description**\
> We agree; having found good performance in diverse cases of time series that are not immediately arising from dynamical systems, is one of the most interesting outcomes of our work.
> Despite our paper would certainly benefit from inspecting such aspects, we argue that it would be better suited for dedicated future work, e.g. compelling follow-ups on time series representation learning and state space reconstruction as a possible underpinning.
> We believe these results, as well as future directions, are of great interest to both dynamical systems and time series communities.
> We thank the reviewer for highlighting this future direction and we are happy to include it in the Conclusions section of our paper.
>
>
> **Q2. What does interpretable refer to in this context?**\
> We kindly refer the reviewer to our common response \#1.
>
>
> We hope to have clarified all of the reviewer’s concerns and are happy to provide further details.
>
> **References:**\
> [a] Bianchi et al., Reservoir Computing Approaches for Representation and Classification of Multivariate Time Series, IEEE Transactions on Neural Networks and Learning Systems, 2020.

---

> ### Comment · Reviewer_RJLc · 2023-08-14
>
> I want to thank the authors for their detailed responses and updated results.
> Overall, The responses and new results cleared up some questions, and there seems to be an overall consent that this paper is a good contribution, so I vote for acceptance.
>
> Some small further comment:
> I agree with one of the referees that section 4.4. (kPCA visualization) relies only a single example, and no more thorough statistical study of kPCA visualisations across datasets is provided.
> The authors claim that "Despite projections being similar for most datasets, we observed a few critical differences" without really making these differences explicit, and it is not clear to me whether this benefit over rmESN really extend to other datasets, or how much this example relied on "cherry-picking".
>
> That's why I would suggest that the authors either make the selection process for the example more explicit/consider other examples and list the "critical differences" explicitly, or move this section to the appendix as an example visualisation, and instead add some of the new results and explanations to the main text.

---

> > ### Author Response · Authors · 2023-08-21
> >
> > We thank the reviewer for their support.
> >
> > We also welcome their suggestion and have moved the kPCA visualization example to the appendix.
> > This additional space in the main text is used to discuss the new results and enhance the discussion along the lines of our common responses #1 and #2.

---

### Author Rebuttal · Authors · 2023-08-08

We would like to remark here our thanks to all reviewers for their extensive reviews.
We are happy to hear that reviewers acknowledged novelty ($\color{red}{\textbf{RJLc}}$, $\color{blue}{\textbf{hHQP}}$) and relevancy ($\color{blue}{\textbf{hHQP}}$, $\color{green}{\textbf{8sG2}}$) of our work, as well as praising its interdisciplinarity ($\color{red}{\textbf{RJLc}}$, $\color{magenta}{\textbf{Nbgs}}$, $\color{orange}{\textbf{XEWY}}$).
We are also grateful to all 5 reviewers for all sharing their appreciation for the clarity of the writing.

In order to strengthen our paper, incentivized by the reviewers' feedback, we have brought sever minor modifications together with the following significant additions:

- we now compare computational time of NVARk* with respect to NVARk and baseline methods, highlighting performance trade-off ($\color{green}{\textbf{8sG2}}$);
- we now provide an additional study of the sensitivity of our approach to the different hyperparameters, either by providing additional experiments (Fig. 1) or extensive discussion ($\color{magenta}{\textbf{Nbgs}}$);
- we now present a novel ablation study (Tab. 1), deepening the importance of linear and non-linear components of NVARk ($\color{magenta}{\textbf{Nbgs}}$);
- we now extend our evaluation to 8 more univariate and 3 more multivariate datasets (Tab. 2) ($\color{orange}{\textbf{XEWY}}$).

Please find below responses which we hope help to clarify shared concerns across more reviewers.
Finally, please find attached a one-page pdf providing the additional experiments that are referenced across our responses.
We look forward to the reviewers' individual responses to our rebuttal.

**1. Discussion of the interpretability claims**\
Our claims refer to NVAR's ability to circumvent the inherent randomness of RC methods.
As in ll.184-186, NVARk representation vectors encapsulate the mutual and auto-mutual information within the dimensions of the input.
In contrast, representations from RC methods encompass hardly-interpretable reservoir dynamics.
Additionally, the key parameters of NVARk are easy to interpret: $s$ controls the spacing between lags and just has to be set high enough not to sample noise; $k$ controls the number of lags and extends memory to find the relevant patterns.
On the other side, for instance, input scaling and spectral radius are much less interpretable.
These are two notable RC parameters controlling, among other aspects, the degree of nonlinearity injected into the reservoir.
How much of this is required by the task is not easy to judge and requires experienced insight into nonlinear dynamics or, most of the time, traditional hyperparameter tuning.
In our view, the absence of heuristics to choose these parameters also supports this.
We have now made this connection between our claims and the above considerations more explicit in Sec. 3.2 of our updated paper.


**2. Limitations and behavior with high-dimensional datasets**\
As the input dimensionality approaches $\bar{d}_r$ (our limit), there is little margin for adding more dimensions.
Here, concatenation of lags and non-linear terms would require substantially increasing $\bar{d}_r$.
This places excessive strain on the linear readout, which inherently possesses limited expressive capacity.
This component struggles to effectively model intricate relationships among numerous features and is increasingly overloaded with spurious correlations.
We then expect the effectiveness of NVARk to decrease.
To overcome this, one can consider replacing random sampling with strategies that prioritize the selection of meaningful terms.
In fact, we believe this is a promising future direction for our work.
Alternatively, it would be interesting to explore how different linear layers, e.g. Lasso, would perform in this regime.
We acknowledge the significance of this aspect in our work, which may not have received sufficient attention in the main text.
We now integrate the provided discussion into the Conclusions of the manuscript.

---

### Decision · Program_Chairs · 2023-09-21

**Decision:**

Accept (poster)

**Comment:**

Five reviewers were assigned to this paper. Four reviewers support acceptance, and a fifth reviewer, Nbgs, supports rejection. Reviewer Nbgs, who supports rejection, did not reply to the authors' rebuttal or get involved in the post-rebuttal discussion. I read Reviewer Nbgs comments, and I did not find any of these comments as a cause for rejection. The comments are mostly in the sense of requesting further clarifications, which the authors have provided in the rebuttal. Therefore, I accept the paper.